# PATEin: A Privacy-Preserving Framework for Knowledge Integration via Adaptive Teacher Selection in C-LLMs

## Abstract

In-context learning (ICL) enables task adaptation without modifying model parameters, making it well-suited for commercial large language models (C-LLMs) with closed-source constraints. However, ICL prompts often contain sensitive information, raising significant privacy concerns. Most existing privacy-preserving methods for ICL require access to model parameters, making them incompatible with C-LLMs. Recent methods based on teacher ensembles with differentially private aggregation have shown promise but face two fundamental challenges: ensemble inconsistency and limited knowledge integration. We propose PATEin, a novel privacy-preserving knowledge transfer framework that dynamically selects the optimal individual teacher model for labeling, thereby mitigating the loss of individual knowledge. Furthermore, it introduces a supervised teacher strategy that selectively incorporates high-consistency voting, effectively integrating individual and ensemble knowledge. Experiments on various C-LLMs (e.g., GPT-3.5-turbo, GPT-4o-mini, Claude-3.5-haiku, DeepSeek-v3) demonstrate that PATEin significantly improves labeling accuracy, reduces computational overhead, and consistently outperforms existing baseline methods.

## 1 Introduction

In-context learning (ICL) has emerged as a powerful means of leveraging LLMs without updating model parameters (Brown et al., 2020; Radford et al., 2019) by embedding task examples directly into prompts (Raffel et al., 2020). However, ICL raises significant privacy concerns, as prompts often contain sensitive and task-specific information (Yu et al., 2024). During inference, LLMs may implicitly leverage this information to better align with user intent, thereby increasing the risk of unintended leakage (Tang et al., 2024; Yu et al., 2024). This risk becomes particularly critical in C-LLMs scenarios, where concurrent user access to a shared model increases the attack surface and amplifies the potential for inference-time privacy breaches. Duan et al. (2023) used prompt-level Membership Inference Attacks (MIA) to effectively infer sensitive information from prompts, thereby exposing fundamental vulnerabilities.

Various methods have been proposed to mitigate these risks. Some aim to reduce prompt exposure while preserving performance (Hu et al., 2024), while others integrate Differential Privacy (DP) (Dwork, 2006) into ICL, generating synthetic inputs aligned with private distributions (Edemacu & Wu, 2025; Flemings et al., 2024; Amin et al., 2024; Wu et al., 2024b; Song et al., 2024). End-to-end frameworks enhance utility and readability by removing contextual private features from prompts (Shen et al., 2024). However, mainstream methods—such as DP-based fine-tuning (e.g., DP-SGD (Abadi et al., 2016)) or model-guided prompt synthesis (Gao et al., 2025)—generally require access to model parameters or gradients, which is infeasible under C-LLM's closed-source constraints.

To address this, PromptPATE (Duan et al., 2023) extends the Private Aggregation of Teacher Ensembles (PATE) framework (Papernot et al., 2018) to prompt-based knowledge transfer. In this setting, C-LLMs act as teacher models to label public data through noisy ensemble voting, producing pseudo-labeled datasets for training student models. PromptPATE delivers strong labeling performance without accessing model internals or private data, ensuring privacy. This separation is further reflected in its architecture: teacher models are trained within private environments, while compact student models

are deployed publicly, enhancing cost efficiency and deployment flexibility. Recent works build upon PromptPATE to explore privacy-preserving prompt synthesis and domain knowledge transfer (Hong et al., 2024a; Li et al., 2024b), enhancing adaptability and scalability while maintaining privacy.

Despite these advances, leveraging C-LLMs for privacy-preserving knowledge transfer still faces two significant challenges: ensemble consistency and knowledge integration. Firstly, model access is limited to black-box API calls due to C-LLMs' closed-source nature (DeepSeekAI, 2023; OpenAI, 2023; Anthropic, 2023). Label generation thus relies on ensemble voting, typically by majority rule. While PromptPATE introduces DP via noisy aggregation (Duan et al., 2023), ensemble consistency remains problematic. Majority voting suppresses divergent yet informative outputs from individual teacher models. Furthermore, restricted API outputs hinder confidence estimation, limiting the ability to leverage each teacher model's unique strengths. Given the time and financial cost of repeated queries, scalability is also a concern. Secondly, current methods lack strategies to combine ensemble and individual teacher models knowledge effectively. While ensemble voting captures aggregate consensus, individual teachers may encode more relevant insights for specific inputs. Replacing ensemble voting with an optimal individual teacher model's output could reduce costs and improve label quality—but identifying such teachers remains challenging. Attempts based on embedding similarity have proven unreliable, and selected teachers often exhibit overconfidence, leading to unstable predictions without supervision.

To address these challenges, we propose PATEin, a novel method that extends the PATE framework (Duan et al., 2023) to enable effective, privacy-preserving knowledge integration and transfer under C-LLM constraints. PATEin targets cost-efficient, privacy-preserving knowledge transfer, enabling student models to approach teacher-level performance on downstream tasks. To resolve ensemble consistency, PATEin selects the most relevant teacher model based on its similarity to the unlabeled public data and delegates labeling accordingly. PATEin preserves individualized knowledge and reduces query overhead. To further integrate ensemble insights, PATEin invokes ensemble voting selectively on high-consistency inputs, incorporating ensemble-level knowledge only when teacher agreement is strong. To our knowledge, PATEin is the first to integrate ensemble-level (via high-consistency voting) jointly and individual-level (via optimal individual teacher model selection) knowledge within the PATE framework in C-LLMs settings. We evaluate PATEin on leading C-LLMs including GPT (OpenAI, 2023), Claude (Anthropic, 2023), and DeepSeek (DeepSeekAI, 2023), benchmarking against four baselines including PromptPATE (Duan et al., 2023). Results demonstrate that PATEin achieves superior labeling accuracy and cost-efficiency under equivalent privacy constraints.

In summary, our main contributions are:

- We propose PATEin, the first privacy-preserving knowledge integration framework tailored to C-LLM settings, addressing ensemble consistency and knowledge integration challenges.
- We identify the loss of individual teacher knowledge in ensemble voting and show that selecting the optimal individual teacher model can be an effective alternative.
- Experiments across multiple C-LLMs demonstrate that PATEin consistently outperforms prior methods in accuracy and cost-efficiency under privacy guarantees.
- We release the code and datasets of PATEin on GitHub(PATEin, 2025) to facilitate reproducibility and future research.

## 2 BACKGROUND AND RELATED WORK

### 2.1 PROMPT IN ICL

ICL enables LLMs to generalize across tasks using a few examples, without parameter updates, leveraging growing model and dataset scales (Yadav, 2024; Liu et al., 2023; Huang et al., 2022; Brown et al., 2020; Radford et al., 2019). Typically presented in a few-shot format, ICL performs inference based on pretraining knowledge (Dong et al., 2024a; Brown et al., 2020). As shown in Figure 1, an ICL prompt for a classification task consists of instruction, labeled example data in [sentence, label] format, and unlabeled public data. The LLM outputs a label result based on token probabilities, but in C-LLMs, only the top-ranked token is accessible. The term "model" (e.g., Teacher or Student) refers to the LLM and its prompt. ICL's structured format and minimal training overhead make it suitable

Figure 1: Overview of the ICL framework for classification task in LLMs.

for complex reasoning (Duan et al., 2023). Central to ICL, prompts can be categorized as soft or discrete (Liu et al., 2022; Lester et al., 2021; Gao et al., 2021). While soft prompts are learnable vectors, their opacity and parameter dependence limit applicability in C-LLMs. In contrast, discrete prompts use interpretable natural language templates without model access, making them ideal for commercial use (Liu et al., 2022; Guo et al., 2022). These prompts can be few-, one-, or zero-shot; as performance varies with example count (Zhao et al., 2021), we adopt a consistent one-shot setting (Duan et al., 2023).

## 2.2 Privacy risks and defense methods in LLMs

LLMs have achieved remarkable success in domains such as healthcare (Huang et al., 2022), aerospace (Yadav, 2024), and chip design (Liu et al., 2023), demonstrating robust performance across diverse downstream tasks (Liu et al., 2025; Li et al., 2024a). LLMs are trained on large-scale datasets that often contain private or third-party data (Jovanović et al., 2025), making them vulnerable to memorization (Duan et al., 2023; Panda et al., 2023) and privacy attacks such as MIA (Maini et al., 2024; Carlini et al., 2022; Shokri et al., 2017), backdoor injection (Chaudhari et al., 2024), and model extraction (Wen et al., 2023). Broader vulnerabilities have been reported under various threat models (Miranda et al., 2025; Maini et al., 2024; Chadha et al., 2024; Li et al., 2024a;d; 2023). In LLMs, user prompts are often logged for alignment (Yu et al., 2024), and ICL may further increase leakage risks as prompts can expose embedded knowledge (Jiang et al., 2020; Davison et al., 2019; Petroni et al., 2019). Duan et al. (2023) demonstrated prompt-level MIA, highlighting privacy risks in interactive settings.

To counter these threats, early defenses focused on redacting sensitive content (Dong et al., 2024b;c), while parameter-based approaches like fine-tuning and federated learning (Sun et al., 2024; Lester et al., 2021; Kairouz et al., 2021) helped reduce exposure. DP techniques such as DP-SGD (Abadi et al., 2016) offer formal guarantees via noise injection, though prompt-level adaptations remain limited (Tang et al., 2025; Wu et al., 2024a; ?). PromptDPSGD (Duan et al., 2023) pioneered DP-SGD for soft prompts, followed by extended prompt-level DP techniques (Gao et al., 2025; Tang et al., 2024; Zhao et al., 2024; Zheng et al., 2024; Amin et al., 2024; Chen et al., 2024; Wu et al., 2024b; Yu et al., 2024). PATEin centers on C-LLMs and fundamentally differs from the aforementioned works. Most assume access to model parameters or data—is unrealistic in API-restricted C-LLMs like GPT (OpenAI, 2023) or Claude (Anthropic, 2023). PromptPATE (Duan et al., 2023) addresses this by extending PATE (Papernot et al., 2018) to discrete prompts, using teacher ensembles to label data privately without exposing inputs. This inspired secure prompt transfer methods (Hong et al., 2024b; Li et al., 2024c), though challenges persist in aggregating ensemble knowledge effectively. Our work tackles this by improving label quality and transfer efficiency in discrete prompt-based frameworks.

## 3 Ensemble consistency challenge

Knowledge transfer allows student models to leverage ensemble knowledge from pre-trained teacher models, thereby enhancing performance on downstream tasks (Papernot et al., 2018). In contrast to open-source LLMs(Brown et al., 2020), C-LLMs (DeepSeekAI, 2023; OpenAI, 2023; Anthropic, 2023) are closed-source and provide limited output parameters, restricting access to individual teacher knowledge. This prioritizes ensemble effects while diminishing individual uniqueness. Additionally, the cost of each C-LLM query impacts the efficiency and feasibility of knowledge transfer (see details in Appendix A).

To safeguard privacy, student models are trained on aggregated knowledge derived from teacher ensemble voting and noise aggregation (Duan et al., 2023), replacing sensitive information (labels)

with noise-aggregated labels. While this method maintains privacy and improves performance, the limited outputs of C-LLMs (e.g., label predictions) and lack of access to complete probability distributions or other crucial data restrict the ability to utilize individual knowledge during teacher ensemble voting effectively.

We study the loss of individual knowledge during teacher labeling. Experimental details are in Appendix A. Figure 2a highlights a key issue: for unlabeled public data in the "Hard" category

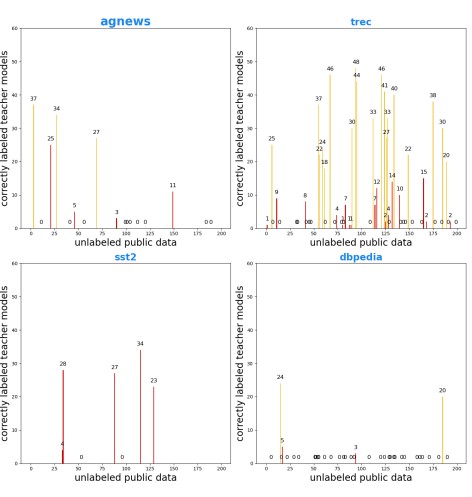
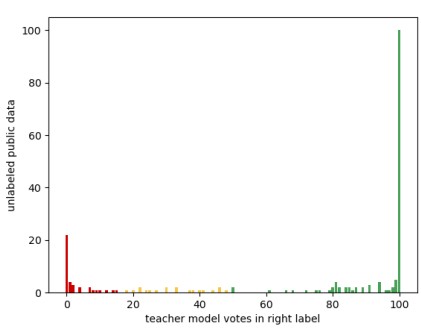

(a) Correct labeling of unlabeled public data.       (b) Correct labeling curve.

Figure 2: Labeling results. Left: Labeling accuracy of 100 teacher models on 200 unlabeled public data, where line color indicates dataset category and line height reflects the number of correct labels. Right: Correct labeling curve on the trec datasets, categorizing samples into "Easy" (green), "Uncertain" (yellow), and "Hard" (red) based on a predefined threshold.

(marked in red), the proportion of teacher models correctly labeling the data falls below the minimum threshold required by Classification standards (details in Appendix A), with some cases showing no correct labels at all. This issue is particularly evident in the dbpedia and trec datasets, where teacher ensemble voting fails to assign correct labels.

Although a substantial number of teacher models correctly label data in the "Uncertain" category (yellow), these labels may still be inaccurate due to the "majority wins" principle in voting or the impact of weight allocation. For example, in the trec dataset, the data point "When was Abraham Lincoln born?" was correctly labeled by only 46 teacher models, while 54 mislabeled it. As a result, the majority's incorrect label determined the final output despite many correct labels, highlighting a loss of valuable individual knowledge.

Further analysis of the trec dataset (Figure 2b) shows the distribution of labeling accuracy. The green bars represent "Easy" data (100% labeling accuracy), the yellow bars represent "Uncertain" data (less than 100% accuracy), and the red bars represent "Hard" data (0% accuracy). The highest peak near the 100 teacher models indicates excellent individual performance, while the second peak near 0 suggests that teacher models struggle with "Hard" data. Additionally, the "Uncertain" category in most datasets had a 0% correct labeling rate, except for trec, where only 10.52% were correctly labeled.

These findings underscore the loss of individual knowledge during the teacher ensemble voting process, limiting the ability to acquire accurate ensemble knowledge.

## 4 KNOWLEDGE INTEGRATION CHALLENGE

Based on Section 3, we identify that knowledge transfer in C-LLMs risks losing individual teacher knowledge and increasing commercial overhead. This section explores the selection of optimal individual teacher models for labeling public data and investigates the integration of individual and

ensemble knowledge to balance cost and accuracy. We focus on: (1) the feasibility of preserving individual knowledge during transfer, and (2) the synergy between individual and ensemble knowledge for improved labeling.

## 4.1 EXPRESSION OF INDIVIDUAL KNOWLEDGE

Figure 2a shows that specific teacher models can accurately label unlabeled public data. This suggests the feasibility of identifying optimal individual teacher models and leveraging their knowledge as an alternative to ensemble voting. However, existing work on knowledge transfer in C-LLMs has not explicitly addressed matching optimal individual teacher models to specific labeling tasks on public datasets (Duan et al., 2023). In practice, differences among teacher models primarily stem from the distinct example data used during their training. Thus, we reformulate the problem by identifying teacher models likely to generate correct labels based on the similarity between unlabeled public data and the internal example data of teacher models, effectively framing it as a text similarity matching task.

While textual similarity often indicates semantic relatedness(Ren et al., 2021), its correlation with labeling accuracy in knowledge transfer is unclear. Specifically: (1) Higher similarity may imply shared knowledge categories, leading to correct labeling; (2) Higher similarity may increase the likelihood of correct predictions; (3) Alternatively, no meaningful correlation may exist, suggesting textual similarity does not reliably predict labeling accuracy.

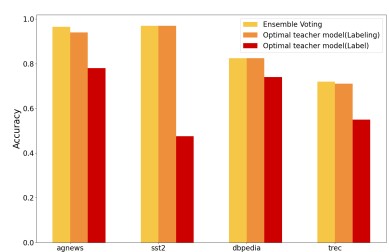

Figure 3: Comparison of labeling accuracies. The figure compares labeling accuracies on the agnews, sst2, dbpedia, and trec datasets, illustrating: (1) the accuracy from ensemble voting among teacher models (gold), (2) the individual accuracy of optimal individual teacher models (orange), and (3) the agreement between optimal individual teacher models and ground-truth labels (red).

We conduct a series of experiments to validate the three proposed scenarios (experiments set up details in Appendix A). Figure 3 compares the labeling accuracies of ensemble voting, optimal individual teacher models, and individual labels against ground-truth labels. Across the four datasets, both ensemble voting and optimal individual teacher models achieve higher accuracy than similarity-based matching. Specifically, ensemble voting outperforms individual labeling on agnews and trec, while on dbpedia and sst2, their performances are comparable. These results demonstrate the feasibility of selecting optimal individual teacher models via similarity matching, though their labeling accuracy still does not surpass that of ensemble voting.

## 4.2 FEASIBILITY OF KNOWLEDGE INTEGRATION

Leveraging the optimal individual teacher model can substantially reduce labeling costs in real-world C-LLM environments, offering substantial potential for commercial applications. However, their standalone labeling accuracy still falls short of that achieved through teacher ensemble voting. Moreover, optimal individual teacher models may exhibit overconfidence as the ensemble size fluctuates, leading to accuracy degradation and instability (discussed in Section 6.3). Thus, individual knowledge cannot fully replace ensemble knowledge at this stage.

To explore the integration of individual and ensemble knowledge, we compared the labeling performance of optimal individual teacher models and ensemble voting across unlabeled public data. Specifically, we randomly sampled 100 public data from the agnews dataset (Section 4.1) and visualized the results in Figure 4. In the figure, dark blue areas indicate public data correctly labeled only by ensemble voting, while light orange areas indicate public data correctly labeled only by optimal individual teacher models; overlapping correct labels are shown with adjusted transparency, and blank areas represent failed cases. The results show that each method successfully labels specific public data that the other fails to label, suggesting their complementary strengths and the potential to integrate individual and ensemble knowledge further to enhance labeling accuracy.

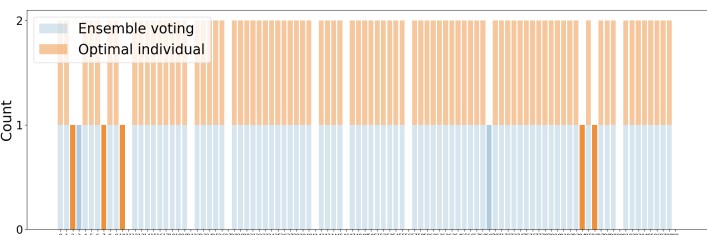

Figure 4: Labeling performance comparison: optimal individual vs ensemble voting.

## 5 PATEin framework

Existing work (Duan et al., 2023) follows the general PATE process of training teacher models, transferring knowledge, and training student models. However, it mainly addresses traditional supervised learning and does not fully tackle ensemble consistency and knowledge integration challenges in C-LLMs. Specifically, the capabilities of optimal individual teacher models are underutilized in ensemble voting, leading to knowledge loss and higher costs.

To address this, we propose PATEin, a framework that ensures optimal teacher selection, reducing cost while maintaining labeling accuracy. Unlike prior methods, PATEin integrates both ensemble and individual knowledge for C-LLM knowledge transfer.

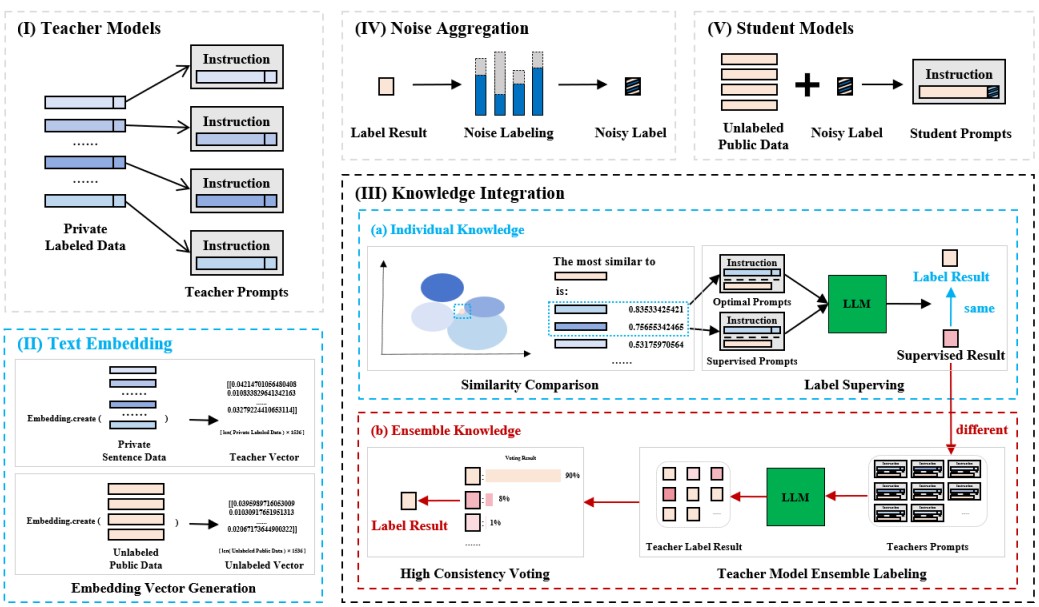

Figure 5: The framework of PATEin.

The process of PATEin is divided into five stages: Teacher Models, Text Embedding, Knowledge Integration, Noise Aggregation, and Student Models, as illustrated in Figure 5.

**Teacher Models.** Each teacher model $T_i$ is trained on disjoint private data $D_i^T = \{(x_k^T, y_k^T)\}$ using one-shot prompting with calibrated templates (Zhao et al., 2021).

**Text Embedding.** Let $N_T$ be the number of teacher models and $N_U$ the number of unlabeled public data samples. To identify the optimal individual teacher model, we convert teacher example data $\{x_i^T\}_{i=1}^{N_T}$ and unlabeled public data $\{x_j^U\}_{j=1}^{N_U}$ into embedding vectors. We employ a hybrid embedding strategy that leverages the complementary strengths of different embedding paradigms. For each input pair $(x_i^T, x_j^U)$, We compute cosine similarity using two embedding models: doc2vec

($\phi_{\text{doc2vec}}$) for document-level topics and text-embedding-3-small ($\phi_{\text{embed3}}$) for contextual features. The similarity for embedding model $\phi_m$ is:

$$S_{ij}^m = \frac{\phi_m(x_i^T) \cdot \phi_m(x_j^U)}{\|\phi_m(x_i^T)\| \cdot \|\phi_m(x_j^U)\|} \tag{1}$$

The final similarity score is the maximum of the two:

$$S_{ij} = \max\left(S_{ij}^{\text{doc2vec}}, S_{ij}^{\text{embed3}}\right) \tag{2}$$

This discrete selection mechanism dynamically chooses the embedding source that yields the highest semantic similarity for each input pair, where $\phi_{\text{doc2vec}}$ captures document-level topical information and $\phi_{\text{embed3}}$ captures fine-grained contextual features. The robustness of this approach to embedding choices is thoroughly analyzed in Appendix G.

**Knowledge Integration.** We construct a similarity matrix $S \in \mathbb{R}^{N_T \times N_U}$. For each $x_j^U$, the optimal individual teacher model $T^*(x_j^U)$ and second-best teacher model $T'(x_j^U)$ are selected:

$$T^*(x_j^U) = \arg\max_i S_{ij}, \quad T'(x_j^U) = \arg\max_{i \neq i^*} S_{ij} \tag{3}$$

Let $y^*$ and $y'$ be their labels. The final label is assigned as:

$$y_j = \begin{cases} y^*, & \text{if } y^* = y' \\ \text{High-consistency voting}(\{T_i(x_j^U)\}_{i=1}^E), & \text{otherwise} \end{cases} \tag{4}$$

Where $E$ denotes the total number of teacher models. This ensures individual knowledge is preserved when teachers agree, while falling back to ensemble voting when uncertain.

**Noise Aggregation.** We apply the Confident-GNMax mechanism to enforce differential privacy (privacy analysis in Appendix F). Let $n_{i,j}(x)$ be an indicator whether teacher $T_i$ assigns label $j$ to public data $x$. The noisy count for label $j$ is:

$$\tilde{n}_j(x) = \sum_{i=1}^E n_{i,j}(x) + \mathcal{N}(0, \sigma^2) \tag{5}$$

where $\mathcal{N}$ denotes Gaussian noise. This ensures $(\varepsilon, \delta)$-DP labeling while preserving compatibility with baseline methods(Duan et al., 2023).

**Student Models.** The noisy labels $\{\tilde{y}_j\}$ are paired with public data $\{x_j^U\}$ to construct student knowledge. These labeled pairs are used to create student prompts for downstream tasks in C-LLMs.

## 6 EXPERIMENTS

In this section, we evaluate PATEin, demonstrating its ability to leverage individual knowledge and integrate ensemble knowledge in cost-effective C-LLM environments. PATEin achieves higher labeling accuracy and facilitates efficient, high-quality knowledge transfer from private teacher models to student models. All experiments were tested on a MacBook Pro Laptop with an 8-core Apple M1 CPU with 16GB RAM.

### 6.1 EXPERIMENTAL SETUP

**Datasets:** To ensure the generalizability of the results, we conduct multi-class labeling tasks on four unlabeled public datasets: agnews, dbpedia, sst2, and trec, as per the baseline in Duan et al. (2023). This setup enables a comprehensive evaluation of the proposed method's effectiveness. **C-LLMs:** Unless stated otherwise, GPT-3.5-turbo is used as the C-LLM for experiments. Section 6.2 introduces additional C-LLMs—Claude-3.5-haiku, GPT-4o-mini, and DeepSeek-v3 for comparison. **Baselines:** In Section 6.2, PATEin is compared against four baseline methods: zero-shot, average performance of individual teacher models, PromptPATE, and the optimal individual teacher model. **High consistency voting threshold:** The default threshold is set to 0.9. Section 6.3 will explore the impact of varying

Table 1: Performance of PATEin.

| Private | Zero-shot | | Single (AVG) | | PromptPATE | | Optimal individual teacher model | | | PATEin | |
|---------|-----------|--------|--------------|--------|------------|--------|----------|------------|--------|------|--------|
| | Acc | Tokens | Acc | Tokens | Acc | Tokens | Acc (Ans) | Acc (Label) | Tokens | Acc | Tokens |
| sst2 | 0.885 | 7419 | 0.9708 | 13900 | 0.975 | 1390032 | 0.45 | 0.965 | 21985 | 0.9847 | 63026 |
| agnews | 0.865 | 16227 | 0.8992 | 27525 | 0.915 | 2752503 | 0.875 | 0.83 | 43356 | 0.9202 | 232996 |
| trec | 0.7 | 9870 | 0.7139 | 13566 | 0.72 | 1356600 | 0.71 | 0.55 | 17228 | 0.7571 | 198043 |
| dbpedia | 0.79 | 24382 | 0.8172 | 39286 | 0.825 | 3928600 | 0.825 | 0.74 | 60569 | 0.8541 | 195950 |
| dbpedia (Claude-3.5-haiku) | 0.2 | 26418 | 0.7165 | 39782 | 0.765 | 3978200 | 0.74 | 0.74 | 61065 | 0.7708 | 315679 |
| dbpedia (GPT-4o-mini) | 0.9 | 23794 | 0.9011 | 39827 | 0.94 | 3982696 | 0.945 | 0.74 | 59511 | 0.9639 | 236054 |
| dbpedia (DeepSeek-v3) | 0.72 | 25982 | 0.7264 | 40868 | 0.74 | 4086780 | 0.74 | 0.73 | 63007 | 0.7552 | 284315 |

this threshold on labeling accuracy. **Prompt template:** To maintain consistency with Duan et al. (2023), we adopt the prompt template from Zhao et al. (2021). **Unlabeled public data:** The unlabeled public dataset contains 200 input sequences from each dataset with known correct labels. There is no overlap between this dataset and the teacher models' example data. **Teacher models:** One hundred teacher models are deployed for each dataset to label the unlabeled public data. Section 6.3 will discuss the effect of varying the number of teacher models. Each teacher model generates label results, which are compared against the correct labels to assess labeling performance across datasets.

## 6.2 OVERALL PERFORMANCE

We compare PATEin against four baselines: (1) Zero-shot, which omits example-specific knowledge; (2) Single (AVG), representing the mean performance of all teacher models; (3) PromptPATE, which aggregates ensemble knowledge via ensemble voting; and (4) Optimal individual teacher model, which selects the optimal teacher model for labeling. The evaluation focuses on labeling accuracy and C-LLM token usage, a proxy for computational cost. Experiments were conducted with 100 teacher models and 200 unlabeled public data to ensure comparability and cost-sensitive evaluation data.

**The labeling accuracy has significantly improved.** Labeling accuracy results, shown in Table 1, demonstrate that PATEin significantly outperforms all four baseline methods across various C-LLMs and datasets, consistently achieving the highest accuracy. PromptPATE benefits from teacher model ensemble voting, surpassing the individual teacher model average. The optimal individual teacher model's performance is unstable: while it underperforms the average on datasets like agnews, it occasionally exceeds PromptPATE's performance (e.g., with GPT-4o-mini). Direct selection of teacher models based on high similarity scores improves accuracy notably. Additional tests with models such as GPT-2 and GPT-babbage-002 reveal that GPT models' zero-shot capabilities have improved over time, with GPT-4o-mini nearing the accuracy of the individual teacher model average.

**The additional cost has significantly decreased.** From a cost perspective, the zero-shot approach minimizes token usage by requiring no example data, while PromptPATE, relying on teacher ensemble voting, incurs the highest token consumption. The optimal individual teacher model reduces token usage by nearly 60 times compared to PromptPATE. PATEin effectively mitigates the cost inefficiency of ensemble voting by integrating the optimal individual teacher model, resulting in token consumption between that of the optimal individual teacher model and PromptPATE. Notably, PATEin achieves significant token savings across all datasets, requiring only about 1/22 of PromptPATE's tokens on the sst2 dataset, 1/20 on dbpedia, 1/12 on agnews, and 1/7 on trec. These results highlight PATEin's ability to effectively balance cost and labeling performance. We further examine the robustness of PATEin under adversarial teacher settings; see Appendix H for details.

**The downstream task effect has significantly enhanced.** We evaluate PATEin on downstream task (detail in Appendix B), showing that it yields superior performance over zero-shot baselines while retaining strong privacy guarantees ($\omega < 0.1$, $\epsilon = 10^{-6}$) and preserving valuable knowledge.

## 6.3 ABLATION EXPERIMENTS

For PATEin, there are two key factors to consider: first, the number of teacher models, and second, the threshold for high-consistency voting.

Table 2: Threshold impact in PATEin.

| High-consistency voting threshold | 0.5 | | 0.55 | | 0.6 | | 0.65 | | 0.7 | | 0.75 | | 0.8 | | 0.85 | | 0.9 | | 0.95 | | 1 | |
|---|---|---|---|---|---|---|---|---|---|---|---|---|---|---|---|---|---|---|---|---|---|---|
| Model accuracy & Loss | Acc | Loss | Acc | Loss | Acc | Loss | Acc | Loss | Acc | Loss | Acc | Loss | Acc | Loss | Acc | Loss | Acc | Loss | Acc | Loss | Acc | Loss |
| Datasets — sst2 | 0.97 | 0 | 0.9798 | 0.01 | 0.9798 | 0.01 | 0.9798 | 0.01 | 0.9848 | 0.015 | 0.9848 | 0.015 | 0.9848 | 0.015 | 0.9848 | 0.015 | 0.9847 | 0.02 | 0.9847 | 0.02 | 0.9847 | 0.02 |
| agnews | 0.905 | 0 | 0.9045 | 0.005 | 0.904 | 0.01 | 0.9082 | 0.02 | 0.9128 | 0.025 | 0.9171 | 0.035 | 0.9171 | 0.035 | 0.9162 | 0.045 | 0.9202 | 0.06 | 0.9198 | 0.065 | 0.9141 | 0.07 |
| dbpedia | 0.825 | 0 | 0.8241 | 0.005 | 0.8283 | 0.01 | 0.8283 | 0.01 | 0.8367 | 0.02 | 0.8367 | 0.02 | 0.8497 | 0.035 | 0.8497 | 0.035 | 0.8541 | 0.04 | 0.8497 | 0.035 | 0.8497 | 0.035 |
| trec | 0.7172 | 0.01 | 0.7194 | 0.02 | 0.7231 | 0.025 | 0.7268 | 0.03 | 0.7292 | 0.04 | 0.7404 | 0.055 | 0.7394 | 0.06 | 0.7527 | 0.09 | 0.7571 | 0.115 | 0.7529 | 0.13 | 0.7572 | 0.135 |

**Impact of the number of teacher models.** We evaluated the impact of varying the number of teacher models (from 20 to 200) on labeling performance with 200 fixed unlabeled public data. As shown in Figure 6, PATEin consistently outperforms the optimal individual teacher model, PromptPATE, and zero-shot baselines across all configurations, maintaining SOTA performance regardless of teacher count. This demonstrates its robustness in integrating individual and ensemble knowledge. While PromptPATE benefits from ensemble consensus, its accuracy plateaus as the teacher count grows. The optimal individual teacher model, which selects labels based on similarity, briefly outperforms PromptPATE (notably at

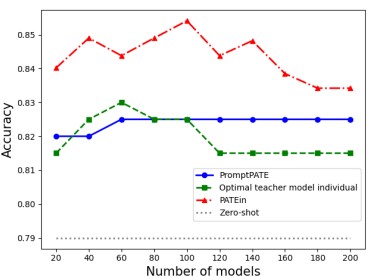

Figure 6: Labeling accuracy trend.

40–60 models) but suffers from overconfidence due to a lack of ensemble correction, leading to a performance decline with more teacher models. PATEin mitigates this by introducing a supervised teacher (second-highest similarity) for self-assessment and using high-consistency voting to correct errors, enhancing reliability. Performance trends show that as the number of teachers increases to 100, both PromptPATE and the optimal individual teacher model improve, boosting PATEin's accuracy. Beyond 100, the growing teacher pool introduces noise and weakens individual influence, slightly reducing PATEin's performance. This highlights a balance point where the integration of ensemble and individual knowledge is most effective.

**Impact of the high-consistency voting threshold.** We further investigate the impact of the high-consistency voting threshold on labeling performance in the high-consistency voting mechanism. As a key design component of PATEin, this threshold critically balances label reliability and data retention. Table 2 presents a comparison of several threshold settings: (1) 0.5—ensemble consensus is fully trusted upon suspected errors, enhancing ensemble knowledge influence; (2) 0.9—the default setting in this study, which selectively discards uncertain samples while retaining reliable ones; (3) 1.0—suspected erroneous samples are discarded without invoking ensemble consensus, favoring precision; and (4) additional intermediate thresholds. Experimental results show that a threshold of 0.9 achieves the best overall performance across four datasets, yielding the highest accuracy on agnews and dbpedia and second-best on sst2 and trec. Notably, fully trusting ensemble votes (0.5) leads to lower accuracy than discarding suspect samples outright (1.0), with a performance gap of 0.04 observed on trec. While lower thresholds preserve more data, they risk admitting noisy labels; higher thresholds improve label quality but reduce usable data. The 0.9 threshold effectively balances this trade-off, maximizing accuracy and data utility. For additional experiments on adversarial robustness, we refer the reader to Appendix H.

## 7 Conclusion and future work

In this paper, we propose PATEin, a novel framework for privacy-preserving knowledge integration and transfer in C-LLM environments. PATEin addresses ensemble inconsistency by selecting the optimal individual teacher model via embedding-based similarity comparison, preserving individual knowledge while minimizing additional costs. To improve knowledge integration, a supervised teacher is introduced to mitigate the overconfidence of the optimal individual teacher model and employ high-consistency voting to fuse individual and ensemble knowledge. Experiments show that PATEin achieves superior labeling accuracy on unlabeled public data at a reduced cost, demonstrating strong commercial potential. Future work will focus on optimizing the teacher selection process, which is critical in effective knowledge integration. We have also explored extending PATEin to long-text generation tasks in C-LLMs (e.g., seqPATE (Tian et al., 2022)), with promising initial results, and will further pursue this direction.

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

## A  EXPERIMENTAL DETAIL

### A.1  DETAIL IN ENSEMBLE CONSISTENCY CHALLENGE

**Experimental setup.   Datasets:** We conduct multi-class classification labeling tasks on unlabeled public data across four different datasets—agnews (Zhang et al., 2015), dbpedia (Zhang et al., 2015), sst2 (Socher et al., 2013), and trec (Voorhees & Tice, 2000) to ensure the results have broad applicability. The agnews dataset consists of news articles categorized into four labels: World, Sports, Business, and Science/Technology. The dbpedia dataset contains structured knowledge extracted from Wikipedia, covering multiple domains such as Company, School, Artist, etc. The sst2 dataset includes movie reviews, primarily used for sentiment analysis tasks, with labels indicating positive or negative sentiment. The trec dataset contains query data for question-answering systems, covering questions such as Number, Location, Person, Entity, Description, etc. **Prompt template:** We follow the baseline work of Duan et al. (2023) and use a standardized prompt template (Zhao et al., 2021). **Unlabeled public data:** We limit the size of the unlabeled public data to 200 input sequences from each dataset. For each sequence, the corresponding correct label is already known. **Classification standards:** We categorize the unlabeled public data into three categories—"Easy", "Uncertain", and "Hard"—based on the performance of individual teacher models on the unlabeled public data. Two thresholds are set for this categorization: Maximum threshold: set to 0.5 by default, this threshold is used to determine if the unlabeled public data belongs to the "Easy" category. If more than 50% of the teacher models correctly label the data, it is classified as "Easy", following the majority rule. Minimum threshold: set by default to 1/number of label types, this threshold is used to further distinguish the "Uncertain" category from the "Hard" category. When fewer than the minimum number of teacher models correctly label the data, it is classified as "Hard". Any unlabeled public data that is not classified as "Easy" or "Hard" is automatically categorized as "Uncertain". **Teacher models:** We use GPT-3.5-turbo as the default C-LLM for all experiments unless otherwise specified, ensuring consistency in the results across different sections. For each dataset, we deploy 100 teacher models. Each teacher model performs an independent labeling task on the unlabeled public data and generates corresponding label results. These label results are then compared to the correct labels to assess the performance of the teacher models in the labeling process across the different datasets.

### A.2  DETAIL IN KNOWLEDGE INTEGRATION CHALLENGE

**Experimental setup.   Datasets:** Following Section 3, we perform multi-class labeling tasks on unlabeled public data using the agnews, dbpedia, sst2, and trec datasets to ensure the generalizability of our results. **Teacher models:** We adopt GPT-3.5-turbo as the default C-LLM for labeling the unlabeled public data across all four datasets. In addition, to account for the potential influence of differences among C-LLMs, we also apply GPT-4o-mini (OpenAI, 2023), Claude-3.5-haiku (Anthropic, 2023), and the recently popular DeepSeek-v3 (DeepSeekAI, 2023) on the dbpedia dataset. For each dataset, we deploy 100 teacher models, each initialized with different example data. **Text embedding model:** To minimize the variability the embedding process introduces, we utilize the state-of-the-art and cost-effective text-embedding-3-small, a lightweight embedding model released by OpenAI (OpenAI, 2023). This model is responsible for converting text data into vector representations. **Similarity comparison algorithm:** In the early stages of the experiment, we tried methods such as Euclidean distance (Danielsson, 1980), Manhattan distance (Chavez et al., 2001), and cosine similarity (Schubert, 2021). Cosine similarity performed the best. Therefore, in this experiment, we use cosine similarity as the metric to determine the similarity between vectors. **Prompt template:** The prompt template used in this section is consistent with that described in Section 3. **Unlabeled public data:** We fix the size of the unlabeled public dataset to 200 input sequences for each dataset, aligned with those used in Section 3.

In this experiment, we first convert the unlabeled public data and the example data within each teacher model into embedding vectors using a text embedding model. This results in vector representations for unlabeled public data and the corresponding example data. Subsequently, we compute the cosine similarity between each unlabeled public data vector and all example data vectors. We identify the example data vector with the highest similarity for each unlabeled public data and record the corresponding match. The teacher model associated with this most similar example is then considered the optimal individual teacher model for that particular instance. For the 200 unlabeled public data, we record the labels assigned by their respective matched optimal individual teacher models and compare

them against the ground-truth labels to compute the labeling accuracy. In parallel, we perform a teacher ensemble voting experiment as a control, using the majority vote among all teacher models to label the same 200 unlabeled public data. The resulting labeling accuracy from this ensemble voting serves as a baseline for evaluating the effectiveness of the optimal individual teacher model.

## B  DOWNSTREAM TASK EXPERIMENTS

To further understand its internal mechanisms, we conduct downstream task experiments focusing on three aspects: (1) the performance of student models trained on example data combining unlabeled public data and PATEin-labeled noisy data; (2) the privacy cost and its trade-off with utility; and (3) the impact of discarding low-consensus ensemble voting on downstream performance.

We selected 50 example data labeled by PATEin (from Section 6.2) to train student models and evaluated them on 100 downstream classification tasks across four datasets. We used the widely adopted C-LLM DeepSeek-v3 (DeepSeekAI, 2023) as the student model and applied the standardized calibration method from Zhao et al. (2021) to control for prompt variability. We compared models trained on "Easy", "Uncertain", and "Hard" samples from the public dataset with those trained on knowledge discarded during high-consistency voting, using the "N\A" label convention to mitigate label noise. Additionally, we benchmarked performance against the teacher and student model zero-shot baselines.

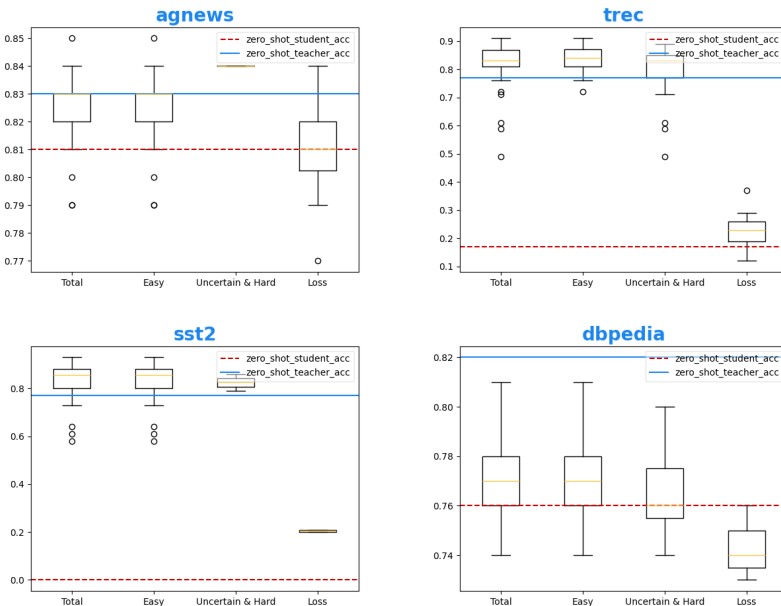

Figure 7: Student model downstream task performance. We selected 50 public data labeled by PATEin to fine-tune the student model, evaluating its performance on 100 downstream classification tasks across four datasets. The label "Total" represents the entire set of unlabeled public data used for training, while "Easy", "Uncertain", and "Hard" refer to subsets categorized by the difficulty of correct labeling, as determined by teacher model performance. The "Loss" category denotes knowledge examples discarded during the high-consistency voting due to insufficient agreement.

**Better downstream labeling performance.**    As shown in Figure 7, the student model trained on the full set of PATEin-labeled public data consistently outperforms its zero-shot counterpart across all datasets and even surpasses the teacher model's zero-shot accuracy on agnews, trec, and sst2. These results highlight effective knowledge transfer, with student models sometimes exceeding teacher performance on downstream tasks.

**Lower privacy cost.** PATEin adopts the privacy-preserving framework from Duan et al. (2023), grounded in secure computation (Choquette-Choo et al., 2021), and is entirely based on the Confident-GNMax algorithm introduced by Papernot et al. (2018). Our analysis demonstrates that PATEin achieves strong DP guarantees ($\omega < 0.1$, $\epsilon = 10^{-6}$), with $\omega$ values of 0.0296 (agnews), 0.0394 (dbpedia), 0.0688 (trec), and 0.01 (sst2). This low privacy cost stems from two key design choices: (1) using an optimal individual teacher model for labeling minimizes privacy leakage, and (2) a high-consistency voting strategy obscures optimal individual teacher model contributions, further enhancing privacy protection. Moreover, PATEin benefits from the post-processing property of DP: once the student model is trained, the privacy guarantee—characterized by ($\omega$ , $\epsilon$)—remains intact regardless of the number or type of queries made to the final prompted C-LLM, providing a strict upper bound on potential leakage associated with the example data.

**Less knowledge loss.** We evaluated student models trained on examples excluded during high-consistency voting to assess the impact of discarded data. As shown in the "Loss" section of Figure 7, these models consistently underperform across all datasets—slightly above zero-shot baselines but well below models trained on other knowledge categories. On dbpedia, performance even regresses below zero-shot. This confirms that PATEin effectively filters low-quality data. Moreover, PATEin significantly boosts performance across "Easy", "Uncertain", and "Hard" categories, leveraging even ambiguous samples—unlike traditional methods that discard them—thereby reducing knowledge loss and preserving valuable information.

## C  IMPACT OF ADDITIONAL COSTS

The classification method for unlabeled public data described in Section 3 is a priori-based. The underlying assumption for accurately classifying unlabeled public data is that the correct labels for each data point are known in advance, which is not the case in real-world environments. However, as illustrated in Figure 2b, existing works (Duan et al., 2023) dealing with unlabeled public data of different categories have employed teacher ensemble voting. By raising the maximum threshold, they aim to select unlabeled public data most likely to belong to the "Easy" category and label them while discarding data from the "Uncertain" and "Hard" categories in order to improve labeling accuracy. Alternatively, by lowering the maximum threshold, they aim to correctly label both the "Easy" and "Uncertain" categories, sacrificing some accuracy in exchange for maximizing the utilization of available knowledge.

In practical C-LLM environments, both approaches—raising or lowering the maximum threshold—can result in significant additional costs. Specifically, most C-LLMs charge based on usage, with fees varying by model and service. For instance, as of February 2025, the cost for GPT-3.5-turbo is \$3.00 per 1 million input tokens (OpenAI, 2025). This means that every million input tokens incur a cost of 3 USD. In the GPT models provided by OpenAI (2023), a token can be a word, part of a word, or even punctuation marks. If teacher ensemble voting is utilized, each teacher model must repeatedly label the same unlabeled public data. This number grows exponentially with the number of teacher models and the amount of unlabeled public data, leading to substantial economic costs. In our experimental process, each prompt contained nearly 300 tokens on average. With 100 teacher models, simply labeling 30 unlabeled public data points using teacher ensemble voting would already cost 3 USD. Additionally, computational resources are consumed each time a teacher model is accessed, leading to significant time costs. In subsequent experiments, we examined the time differences across various C-LLMs when processing the same prompt. Among them, the fastest model was GPT-4o, with an average processing time of less than 1 second, while the slowest was Claude-3.5-haiku, which reached a maximum processing time of up to 12 seconds. These time differences will be further amplified as the number of teacher models and the volume of unlabeled public data increase.

## D  ETHIC AND BROADER IMPACT STATEMENTS

This study introduces a privacy-preserving framework for knowledge transfer from C-LLMs without requiring access to their internal parameters. PATEin can benefit sensitive domains such as healthcare and finance by mitigating prompt-level privacy risks while enhancing downstream task performance. All experiments in this study were conducted using publicly available datasets. Nonetheless, extracting knowledge from C-LLMs may raise concerns related to intellectual property and potential misuse in

# E    LIMITATIONS

First, our current work focuses primarily on downstream classification tasks, which limits its applicability to relatively narrow scenarios. In practice, C-LLMs are predominantly utilized in generative tasks. To this end, we have made preliminary attempts to extend the knowledge integration capability of PATEin to long-form text generation tasks, as in seqPATE(Tian et al., 2022), and have obtained promising initial results. Second, although our approach achieves SOTA performance across benchmarks, the performance margin is not substantially large. This is likely because the quality of knowledge integration is closely tied to the choice of optimal individual teacher models. We employ a basic strategy for identifying optimal individual teacher models, leaving room for improvement. Third, we observe that the high-consistency voting mechanism in PATEin results in discarding some public data. While these discarded labels are less reliable, they exhibit competitive performance in downstream tasks. We plan to investigate this phenomenon in depth to understand the relationship better and leverage such data to construct more effective prompts for student model training. Lastly, we acknowledge that, in our challenge analysis (Section 3 and Section 4), we intentionally adopted majority-vote labels as proxies for denoised probability distributions. This design choice facilitated an intuitive understanding of the observed phenomena. Nonetheless, this simplification falls short of capturing real-world data's full complexity and noise characteristics.

# F    PRIVACY ANALYSIS

While PromptPATE(Papernot et al., 2018) successfully introduced DP to prompt-based learning via the Confident-GNMax aggregator, its reliance on ensemble voting for every query leads to consistency challenges and unnecessary noise injection. To address these limitations, PATEin enhances this foundation with an adaptive teacher selection mechanism, improving utility while maintaining the same rigorous DP guarantees.

## F.1    CORE DP MECHANISM: THE CONFIDENT-GNMAX AGGREGATOR.

PATEin inherits the formal $(\varepsilon, \delta)$-DP guarantees from PromptPATE by utilizing the same underlying Confident-GNMax algorithm during the noise aggregation stage. This mechanism ensures that the act of labeling public data does not leak information about the private training data of the teacher models. The algorithm operates as follows:

---

**Algorithm 1** Confident-GNMax Aggregator in Papernot et al. (2018)

---

**Require:** Input sample $s$, threshold $T$, noise parameters $\sigma_1, \sigma_2$

1: **if** $\max_j \left\{ \sum_{i \in [E]} n_{i,j}(s) \right\} + \mathcal{N}(0, \sigma_1^2) \geq T$ **then**

2:     **return** $\arg\max_j \left\{ \sum_{i \in [E]} n_{i,j}(s) + \mathcal{N}(0, \sigma_2^2) \right\}$

3: **else**

4:     **return** $\perp$                                            $\triangleright$ Abstain if no confident consensus

5: **end if**

---

In PATEin, we adopt this mechanism to aggregate teacher votes only when the consensus is sufficiently high. The noisy vote count for class $j$ is computed as:

$$\tilde{n}_j(x) = \sum_{i=1}^{E} n_{i,j}(x) + \mathcal{N}(0, \sigma^2) \quad \text{(same as Eq. 5)} \tag{6}$$

and the final label is selected only if the confidence threshold is met.

## F.2 Novel mechanisms: adaptive teacher selection and high-consistency voting.

PATEin introduces two key enhancements over PromptPATE: **Adaptive teacher selection:** For each public example data, we identify the top two most similar teacher models based on embedding similarity. **High-consistency voting:** If these teacher models agree, their consensus label is adopted directly; otherwise, we fall back to the DP-protected Confident-GNMax vote.

Critically, our adaptive selection and voting mechanisms introduce no additional privacy cost, as they operate purely as post-processing steps on teacher outputs, without accessing private data. The fundamental property of differential privacy is that any function or analysis applied to the output of a DP mechanism remains differentially private without degrading the original $(\varepsilon, \delta)$ guarantee. The adaptive routing mechanism only interacts with the teacher models through their final output labels. Furthermore, its selection logic operates without any direct access to the original private training data. Therefore, it can safely inherit the rigorous privacy guarantees provided by the underlying Confident-GNMax queries that it relies upon when needed.

**Formal statement.** Formally, if each Confident-GNMax call $\mathcal{A}_t$ provides Rényi DP with parameter $\varepsilon_t(\alpha)$ for order $\alpha > 1$ (Mironov, 2017), then by RDP composition the total privacy cost is

$$\varepsilon_{\mathrm{RDP}}(\alpha) = \sum_{t \in \mathcal{T}} \varepsilon_t(\alpha), \tag{7}$$

where $\mathcal{T}$ is the set of queries requiring aggregation. Converting RDP to $(\varepsilon, \delta)$ guarantees (standard conversion (Mironov, 2017)) gives the overall budget. Since adaptive teacher selection and high-consistency voting are post-processing of teacher outputs, they incur no additional privacy cost beyond these Confident-GNMax invocations.

## F.3 Improved privacy-utility trade-off.

By strategically reducing unnecessary reliance on noisy aggregation, PATEin achieves a stronger privacy–utility trade-off. In practice, 30–50% of queries are resolved by agreement between the two most similar teachers, avoiding noisy aggregation (see Section 6.3 for detailed statistics). Such agreements are treated as high-confidence votes and do not require additional noise injection beyond the Confident-GNMax mechanism. This design increases the proportion of accurate labels available to the student model while ensuring the overall process remains within the strict privacy budget. Consequently, PATEin provides higher labeling accuracy and utility compared to PromptPATE, which must apply noisy aggregation to every query.

## F.4 Experimental privacy and performance validation.

In our experiments, PATEin consistently matched or surpassed the strong privacy parameters of PromptPATE while achieving significantly higher accuracy across multiple datasets and C-LLMs (e.g., GPT-3.5-turbo, Claude-3.5-haiku). The abstention rates $\omega$ remained low (between 0.01 and 0.07), indicating that the framework maintains high utility even under these stringent privacy constraints. This demonstrates that PATEin successfully leverages post-processing strategies to enhance the efficiency and effectiveness of privacy-preserving knowledge transfer, making it a robust solution for real-world C-LLM deployments where both accuracy and privacy are paramount.

# G Sensitivity analysis of embedding choice

The effectiveness of PATEin's teacher selection mechanism relies on measuring semantic similarity between unlabeled public data and the teachers' example data. To assess robustness, we conducted a comprehensive analysis of its sensitivity to different text embedding models and similarity metrics.

**Experimental setup.** We evaluated four embedding strategies on the dbpedia dataset, using 100 teacher models and 200 unlabeled public data, to assess their impact on labeling accuracy: **text-embedding-3-small:** A cost-effective and efficient embedding model from OpenAI (2023). **text-embedding-3-large:** A larger and more powerful embedding model from OpenAI (2023), but also more expensive. **Sentence-BERT:** A widely adopted model for generating sentence-level

embeddings. **Hybrid Strategy (ours):** A discrete selection mechanism that dynamically switches between doc2vec (which captures document-level topics) and text-embedding-3-small (which captures contextual semantics). For each input, the embedding source that yields the higher cosine similarity between an unlabeled public data sample and a teacher example data point is selected.

**Results and conclusion.** The results, summarized in Table 3, show that PATEin maintains stable labeling accuracy across all embedding strategies, with performance variation confined to within $\sim$2.5%. This bounded variation highlights the robustness of our teacher selection framework.

Overall, the more expensive text-embedding-3-large achieved only marginal improvements (e.g., +0.7% on GPT-3.5-turbo relative to our hybrid strategy), which may not justify its higher computational cost. In contrast, our proposed hybrid strategy delivered consistently strong and stable performance across all four C-LLMs, and particularly excelled on non-OpenAI models (Claude-3.5-haiku and DeepSeek-v3), underscoring its superior generalizability. By comparison, Sentence-BERT exhibited more noticeable drops in certain configurations (e.g., –6.8% on GPT-3.5-turbo), suggesting that its embedding space is less aligned with prompt-based tasks in our setting.

We further examined the sensitivity to similarity metrics. Alternatives such as Euclidean and Manhattan distances consistently underperformed relative to cosine similarity, which proved to be the most effective and was therefore adopted as the default.

In summary, this analysis demonstrates that PATEin's performance is not critically dependent on any single embedding model. The hybrid strategy offers a favorable balance between performance, cost-efficiency, and model-agnostic generalizability. Future work may extend this approach through adaptive embedding selection from a broader model pool.

Table 3: Labeling accuracy on dbpedia with different embedding strategies.

| Embedding strategy | GPT-3.5-turbo | GPT-4o-mini | Claude-3.5-haiku | DeepSeek-v3 |
|---|---|---|---|---|
| hybrid strategy | 0.8541 | 0.9639 | 0.7708 | 0.7552 |
| text-embedding-3-small | 0.8467 | 0.9612 | 0.7432 | 0.7168 |
| text-embedding-3-large | 0.8617 | 0.9650 | 0.7641 | 0.7452 |
| Sentence-BERT | 0.7862 | 0.9481 | 0.7600 | 0.7512 |

## H  ROBUSTNESS TO ADVERSARIAL TEACHERS

To further examine the robustness of PATEin, we conducted experiments with adversarial teachers, motivated by the concern that a teacher may intentionally output corrupted labels (e.g., always predicting class 1 regardless of the input). Our consistency-based fallback strategy mitigates such risks. Specifically, a label from the optimal individual teacher model is adopted only when it agrees with the supervised teacher model. Otherwise, we resort to the Confident-GNMax aggregation over the full ensemble with high-consistency voting. This mechanism detects adversarial behavior through disagreement, without requiring explicit adversary identification.

We evaluated PATEin on four datasets (agnews, dbpedia, sst2, trec) while progressively replacing up to 50% of teachers with adversarial ones that always output class 1. We compared three methods: PATEin(selection + adaptive fallback), Individual Selection (optimal indivudual model only, no fallback), and PromptPATE (full ensemble voting, no adaptivity). For each method, we report Label acc (accuracy), Adv (Adversarial) selection rate, and Fallback rate.

Table 4 presents the results on dbpedia, with other datasets showing similar trends. As the proportion of adversarial teachers increases, Individual selection suffers significant degradation (from 82.5% to 47.5% accuracy), while PromptPATE collapses entirely at 50% adversaries (7.5% accuracy). In contrast, PATEin maintains substantially higher robustness (55.2% accuracy at 50% adversaries) by adaptively triggering fallback when teacher disagreement is detected. This confirms that our mechanism effectively mitigates adversarial influence without explicit adversary detection. For completeness, the full experimental results across all datasets are available in our supplementary repository at GitHub (PATEin, 2025).

Table 4: Robustness on dbpedia under adversarial teachers.

| Method | Label Acc (%) | Adv select rate (%) | Fallback rate (%) |
|---|---|---|---|
| Individual Selection | $82.5 \rightarrow 47.5$ | $0 \rightarrow 48.5$ | N/A |
| PromptPATE | $82.5 \rightarrow 7.5$ | N/A | N/A |
| PATEin | $84.9 \rightarrow 55.2$ | $0 \rightarrow 48.5$ | $0 \rightarrow 50.5$ |

# I  LLM USAGE

We report the role of LLMs in the preparation of this paper. Since our research focuses on privacy-preserving frameworks for LLMs, their use as experimental subjects was unavoidable. Beyond this, LLMs were used solely as a general-purpose writing assistant. Specifically, LLMs were employed to check grammar and improve the fluency of English writing. LLMs were not involved in research ideation, algorithm design, experimental execution, or result analysis. All technical contributions, models, experiments, and conclusions in this work are solely by the authors.

