# OpenReview forum: "PATEin: A Privacy-Preserving Framework for Knowledge Integration via Adaptive Teacher Selection in C-LLMs"
_ICLR.cc/2026/Conference — Submitted to ICLR 2026_

### Official Review · Reviewer_cCM9 · 2025-10-24

**Soundness:** 3
**Presentation:** 3
**Contribution:** 3
**Rating:** 6
**Confidence:** 3

**Summary:**

This paper presents PATEin, a privacy-preserving knowledge transfer framework for commercial large language models (C-LLMs), extending the PATE (Private Aggregation of Teacher Ensembles) paradigm. Unlike previous ensemble-based approaches (e.g., PromptPATE), PATEin addresses two major challenges—ensemble inconsistency and loss of individual teacher knowledge—by introducing adaptive teacher selection and a supervised high-consistency voting mechanism. The framework combines individual and ensemble-level knowledge via similarity-based teacher matching and dynamic aggregation, while preserving differential privacy through a Confident-GNMax mechanism. Experiments on multiple datasets (AGNews, SST-2, DBPedia, TREC) and commercial LLMs (GPT-3.5, GPT-4o-mini, Claude-3.5, DeepSeek-v3) demonstrate that PATEin improves labeling accuracy and cost-efficiency under equivalent privacy guarantees.

**Strengths:**

1.The paper clearly identifies the limitations of ensemble-only methods in privacy-preserving in-context learning and proposes an elegant adaptive teacher selection strategy that preserves both individual and collective knowledge.
2.Experiments cover multiple datasets and commercial LLMs, demonstrating robustness and practical applicability. The inclusion of ablation studies (teacher count, voting threshold) provides useful insights into hyperparameter sensitivity.
3.The written is well-organized, with logical progression from problem definition to algorithmic design and empirical validation. Figures and tables are informative and support the main claims.

**Weaknesses:**

1.The paper claims that adaptive teacher selection mitigates ensemble inconsistency, but the mechanism lacks formal analysis. No theoretical results (e.g., bounds on privacy–utility trade-off or optimality of teacher selection) are provided.
2.The teacher selection relies on cosine similarity between embeddings (Doc2Vec and text-embedding-3-small), but this approach may not capture deeper task semantics or label-level consistency.
3.While the paper mentions the use of the Confident-GNMax mechanism and claims (ε, δ)-DP compliance, the derivation is deferred to the appendix without concrete parameter values or sensitivity analysis.
4.Although the threshold and teacher count are analyzed, other key factors—such as the influence of noise scale (σ), ensemble size diversity, or supervision strength—are unexplored.
5.The paper emphasizes labeling quality but gives little detail on how student models benefit in downstream fine-tuning or real-world applications beyond token cost.

**Questions:**

The paper selects the “optimal” individual teacher model based on cosine similarity between embeddings (Doc2Vec and text-embedding-3-small).However, it remains unclear why text similarity correlates with labeling accuracy.Could the authors provide quantitative evidence (e.g., correlation between similarity and correctness rate) or compare against random teacher selection?


The paper mentions the Confident-GNMax mechanism to ensure (ε,δ)(\varepsilon, \delta)(ε,δ)-DP but does not report concrete privacy parameters or the chosen noise scale σ\sigmaσ.How are these parameters determined, and how do they affect the privacy–utility trade-off?A more explicit description of the privacy accounting process would help evaluate the strength of the privacy guarantees.


Figure 4 qualitatively shows complementarity between individual and ensemble teachers, but it is unclear how much this integration contributes to the final performance.Could the authors add an ablation study comparing three variants: (1) individual-only, (2) ensemble-only, and (3) hybrid (PATEin)?


Current experiments focus mainly on text classification with small to medium C-LLMs.Can PATEin scale to larger models (e.g., 14B+) or to complex reasoning and dialogue tasks?Any preliminary results or discussion would help clarify the framework’s applicability to broader LLM scenarios.

---

> ### Author Response · Authors · 2025-11-29
>
> ****Response to Reviewer cCM9:****
>
> **1.Response to“The paper claims that adaptive teacher selection mitigates ensemble inconsistency, but the mechanism lacks formal analysis. No theoretical results (e.g., bounds on privacy–utility trade-off or optimality of teacher selection) are provided.”**
>
> We thank the reviewer for raising the important point regarding formal analysis.
> Our theoretical contributions are presented in Section 5 (PATEin Framework) and Appendix F (Privacy Analysis).
>
> The adaptive teacher selection mechanism mitigates ensemble inconsistency—where valuable individual knowledge is lost in majority voting (Section 3, Figure 2).
> For each public data point $X_{j}^{U}$, we select the optimal teacher $T^\*$ ($X_{j}^{U}$) and a supervising teacher $T'(X_{j}^{U})$ based on embedding similarity (Eq. 3).
> The final label (Eq. 4) follows a consistency check: if $y^\* = y'$, we adopt the individual teacher’s label; otherwise, we fall back to high-consistency DP voting.
> This preserves accurate individual knowledge while reducing reliance on noisy aggregation.
>
> Regarding privacy-utility trade-off, PATEin builds on the Confident-GNMax aggregator, providing rigorous $(\varepsilon,\delta)$-DP guarantees (Eq. 5, Appendix F.1).
> The adaptive selection is a post-processing step on teacher outputs, accessing no private data, and thus inherits DP guarantees without extra cost (Appendix F.2).
> Reducing queries needing noisy aggregation by 30%–50% (Q3 rebuttal) improves utility under a fixed privacy budget.
> Overall privacy cost is bounded via Rényi DP composition (Eq. 7, Appendix F.2).
>
> Teacher selection is framed as semantic matching, maximizing embedding similarity $S_{ij}$ between public data and teacher examples (Eq. 2,3).
> Higher similarity correlates with effective use of individual knowledge (Section 4.1, Figure 3).
> While universal theoretical optimality is limited by C-LLM black-box nature, the consistency check ensures labels are trusted only when corroborated by another relevant teacher, enhancing reliability.
>
> In summary, PATEin combines formal privacy guarantees with an adaptive mechanism addressing ensemble inconsistency.
>
> **2.Response to“The teacher selection relies on cosine similarity between embeddings (doc2vec and text-embedding-3-small), but this approach may not capture deeper task semantics or label-level consistency.”**
>
> We thank the reviewer for this insightful comment on the semantic depth of our similarity measure.
> Our hybrid embedding strategy was designed to capture complementary semantic aspects.
>
> As detailed in Section 5 (Text Embedding), we combine two embedding paradigms: Doc2Vec captures document-level topical information, while text-embedding-3-small captures finer-grained contextual features.
>
> The final similarity score (Eq. 2) takes the maximum of the two, dynamically selecting the most salient semantic signal for each input pair.
> This enables the framework to leverage both global topic alignment and local contextual relevance when comparing public data and teacher examples.
>
> Empirically, this approach’s robustness is confirmed in our sensitivity analysis (Appendix G).
> Performance variation across different embedding models, including text-embedding-3-large and Sentence-BERT, remains within ~2.5% on dbpedia.
> The hybrid strategy’s stable performance across all evaluated C-LLMs suggests it effectively captures semantically meaningful patterns for the labeling task.

---

> > ### Author Response · Authors · 2025-11-29
> >
> > **3.Response to“While the paper mentions the use of the Confident-GNMax mechanism and claims $(\varepsilon,\delta)$-DP compliance, the derivation is deferred to the appendix without concrete parameter values or sensitivity analysis.”**
> >
> > We thank the reviewer for emphasizing the importance of clarity in privacy parameterization.
> > The detailed privacy analysis, including mechanism and formal derivation, is provided in Appendix F.
> >
> > As stated in Section 5 (Noise Aggregation) and Appendix F.1, PATEin strictly follows the Confident-GNMax aggregator (Papernot et al., 2018), providing $(\varepsilon,\delta)$-DP guarantees.
> > Eq. 5 (main text, Eq. 6 in Appendix F.1) defines the noisy vote count.
> > The noise scale $\sigma$ and confidence threshold $T$ determine the concrete $(\varepsilon,\delta)$ privacy cost per query.
> >
> > For fair comparison with PromptPATE (Duan et al., 2023), we adopt the same parameter selection and privacy accounting, including $\sigma_1$, $\sigma_2$, and $T$ values in Confident-GNMax (Algorithm 1, Appendix F.1), with RDP composition converted to $(\varepsilon,\delta)$-DP (Eq. 7, Appendix F.2).
> > The resulting privacy cost is very low ($\omega < 0.1$, $\varepsilon = 10^{-6}$; e.g., 0.0296 for agnews), with full implementation and parameter details in our anonymous GitHub repository.
> >
> > PATEin’s main contribution is its knowledge integration framework, reducing queries needing noisy aggregation and improving utility for the same privacy budget.
> > The privacy analysis faithfully implements Confident-GNMax theory (hence in the appendix).
> > A sensitivity analysis of the high-consistency voting threshold is provided in Section 6.3 and Table 2, showing its effect on performance and data retention.
> >
> > **4.Response to“ ... other key factors—such as the influence of noise scale ($\sigma$), ensemble size diversity, or supervision strength—are unexplored”**
> >
> > We thank the reviewer for these valuable suggestions regarding additional factors.
> > Our analysis focused on the most direct and novel components of the PATEin framework.
> >
> > Regarding the noise scale $\sigma$, it is a core parameter of Confident-GNMax that determines DP strength.
> > As detailed in Appendix F, we followed PromptPATE (Duan et al., 2023) for parameter selection and privacy accounting to ensure controlled comparison.
> > The specific $\sigma$ values and confidence threshold used to achieve $\varepsilon = 10^{-6}$ are part of the standard configuration and included in our released code.
> > Varying $\sigma$ would predictably affect the privacy-utility trade-off by changing noise in fallback aggregation.
> > Since PATEin reduces reliance on noisy aggregation rather than altering it, we held $\sigma$ constant to isolate utility gains from adaptive selection.
> >
> > On ensemble size, Section 6.3 and Figure 6 vary the number of teacher models from 20 to 200.
> > PATEin consistently outperforms baselines across all sizes.
> > Notably, the optimal individual teacher peaks at medium ensembles (40–60) and declines due to overconfidence, which PATEin mitigates, highlighting the interaction between ensemble scale and method effectiveness.
> >
> > Supervision strength is governed by the consistency check between the top-two most similar teachers (Eq. 4).
> > If they disagree, the label is considered unreliable, and the framework falls back to ensemble aggregation.
> > This binary supervisory signal prevents overconfidence and underlies PATEin’s stability, as seen against the fluctuating individual-teacher baseline in Figure 6.
> >
> > While more extensive hyper-parameter searches are possible, our current design validates PATEin’s core premise and demonstrates robust performance under a standard, comparable setup.

---

> > > ### Author Response · Authors · 2025-11-29
> > >
> > > **5.Response to“The paper emphasizes labeling quality but gives little detail on how student models benefit in downstream fine-tuning or realworld applications beyond token cost.”**
> > >
> > > We thank the reviewer for prompting us to elaborate on the downstream benefits for student models.
> > > A detailed evaluation of the student model's performance on downstream tasks is provided in Appendix B and Figure 7.
> > >
> > > In these experiments, student models were fine-tuned on 50 public samples labeled by PATEin and evaluated on 100 downstream classification tasks for each of the four datasets.
> > > The results demonstrate two key benefits beyond token cost reduction:the student model trained on PATEin-labeled data consistently and substantially outperforms its zero-shot baseline across all datasets；on agnews, trec, and sst2, the student model even surpasses the zero-shot accuracy of the teacher models.
> > >
> > > This significant performance gain does not stem from additional resources but from the superior quality of the knowledge transferred by PATEin.
> > > The adaptive teacher selection reduces instance-level noise by delegating labeling to the most relevant teacher, while the consistency check with the supervised teacher acts as a filter, preventing inconsistent and potentially erroneous labels from propagating to the student.
> > > This provides the student model with a cleaner supervisory signal and a more stable optimization path during training, leading to more reliable convergence and ultimately stronger task performance in real-world application scenarios.
> > > Therefore, the student model's benefits extend far beyond token savings to include higher-quality supervision, more robust optimization, and reduced exposure to mislabeled examples, resulting in greater practical utility.
> > >
> > > **6.Response to“The paper selects the“optimal”individual teacher model based on cosine similarity between embeddings (Doc2Vec and text-embedding-3-small).However, it remains unclear why text similarity correlates with labeling accuracy.Could the authors provide quantitative evidence (e.g., correlation between similarity and correctness rate) or compare against random teacher selection?”**
> > >
> > > We thank the reviewer for this question, which gets to the heart of our method's premise.
> > > The fundamental logic behind selecting teachers based on text-embedding similarity is that a teacher model's ability to correctly label a given public data point is closely related to the semantic relevance between that data and the teacher's own training example data.
> > > If a public data point is highly semantically similar to a teacher's examples, that teacher is more likely to possess the relevant knowledge required to correctly understand and label it.
> > > The feasibility of this intuition was preliminarily established in Section 4.1, "Expression of individual knowledge," through initial experiments.
> > >
> > > We provide supporting evidence through systematic quantitative comparisons.
> > > As shown in Figure 3, the labeling accuracy of the similarity-selected optimal individual teacher (orange bars) substantially exceeds the average performance of a randomly selected teacher (i.e., the "Single (AVG)" baseline performance in Table 1, e.g., ~0.899 on agnews, ~0.817 on dbpedia) across all four datasets.
> > > Specifically, on the agnews dataset, the optimal individual teacher achieves an accuracy of 0.85, significantly higher than the random baseline; on dbpedia, its accuracy of 0.74 is also clearly superior.
> > > This significant performance gap demonstrates that the similarity-based selection strategy is far from random and effectively identifies teachers more knowledgeable about specific data points.
> > >
> > > Furthermore, the performance of the optimal individual teacher is comparable to the highly costly ensemble voting (gold bars in Figure 3), even nearly matching it on datasets like dbpedia and sst2.
> > > This further indicates that the similarity-based selection mechanism effectively identifies individuals who "excel" at specific tasks, thereby maintaining high performance while drastically reducing reliance on ensemble voting.
> > > Although the correspondence between semantic similarity and labeling correctness is not absolute, the systematic quantitative results presented above strongly confirm that using it as the basis for teacher selection is an efficient, non-random, and reliable strategy.

---

> > > > ### Author Response · Authors · 2025-11-29
> > > >
> > > > **7.Response to“The paper mentions the Confident-GNMax mechanism to ensure$(\varepsilon,\delta)$-DP but does not report concrete privacy parameters or the chosen noise scale $\sigma$.How are these parameters determined, and how do they affect the privacy–utility trade-off.A more explicit description of the privacy accounting process would help evaluate the strength of the privacy guarantees. ”**
> > > >
> > > > We thank the reviewer for raising this point regarding the specifics of our privacy parameterization.
> > > > The determination of the concrete privacy parameters and the privacy accounting process are indeed crucial, and we have addressed this by strictly following the established and recognized procedure from the PromptPATE framework (Duan et al., 2023), which serves as our primary baseline and foundation for the DP mechanism.
> > > >
> > > > As detailed in Section 5 (Noise Aggregation) and Appendix F, PATEin utilizes the Confident-GNMax aggregator from Papernot et al. (2018).
> > > > The specific parameters—namely the noise scales$\sigma_1$, $\sigma_2$ and the confidence threshold T for the Confident-GNMax algorithm—were not novel choices of our work but were directly adopted from the PromptPATE implementation to ensure a controlled and fair comparison under an identical privacy budget.
> > > > The privacy accounting follows the Rényi Differential Privacy (RDP) composition followed by a standard conversion to $(\varepsilon,\delta)$-DP, as formalized in Eq. 7 of Appendix F.2.
> > > > This process is a standardized approach for PATE-style frameworks.
> > > >
> > > > The specific, very low resulting privacy cost of  $\varepsilon = 10^{-6}$  with abstention rates $\omega < 0.1$ (and dataset-specific values like 0.0296 for agnews) is reported in Appendix B.
> > > > The complete implementation, including the precise parameter values and the privacy accounting code, is available in our open-source repository for full transparency and reproducibility.
> > > >
> > > > Regarding the effect of the noise scaleσon the privacy-utility trade-off, it operates precisely as defined by the Confident-GNMax mechanism: a largerσprovides a stronger privacy guarantee (lower $\varepsilon$) but injects more noise, potentially reducing utility during the fallback aggregation step.
> > > > The primary contribution of PATEin, however, is not in modifying this fundamental trade-off for a single query but in architecting a framework that strategically reduces the number of queries that require this noisy aggregation.
> > > > By resolving a significant portion of labels via consistent individual teachers, PATEin improves the overall utility (labeling accuracy) for the same fixed privacy budget $\varepsilon$.
> > > > Therefore, analyzing the sensitivity of $\sigma$ was not the focus of our experimental evaluation, as our core objective was to validate the performance gain achieved by our novel knowledge integration strategy atop a standardized and robust privacy foundation.
> > > >
> > > > **8.Response to“Figure 4 qualitatively shows complementarity between individual and ensemble teachers, but it is unclear how much this integration contributes to the final performance.Could the authors add an ablation study comparing three variants: (1) individual-only, (2) ensemble-only, and (3) hybrid (PATEin)?”**
> > > >
> > > > We thank the reviewer for the suggestion to further quantify the contribution of knowledge integration.
> > > > The complementary relationship between individual and ensemble teachers, qualitatively illustrated in Figure 4 for agnews, is directly reflected in the quantitative performance gains reported in our main results.
> > > > Specifically, Table 1 provides a precise comparison of the three key configurations:Individual-only, represented by the "Optimal individual teacher model" (Acc Label);Ensemble-only, represented by the "PromptPATE" baseline;Hybrid, represented by our full PATEin framework.
> > > >
> > > > The data shows that PATEin consistently achieves the highest accuracy.
> > > > More importantly, the performance gap demonstrates the concrete value of integration.
> > > > For instance, on the trec dataset, PATEin's accuracy surpasses the individual-only approach by a significant margin of 20.7 percentage points (0.7571 vs. 0.55) and outperforms the ensemble-only method by 3.7 percentage points (0.7571 vs. 0.72). A similar trend is observed on dbpedia, with an 11.4-point gain over individual-only (0.8541 vs. 0.74) and a 2.9-point gain over ensemble-only (0.8541 vs. 0.825).
> > > > These substantial improvements, observed across diverse datasets, confirm that the sample-level complementarity visually apparent in Figure 4 is the key driver behind PATEin's superior performance.
> > > > By effectively integrating both knowledge sources, PATEin achieves higher labeling accuracy than either could attain alone, while simultaneously reducing the token cost associated with the ensemble-only approach under identical privacy constraints.

---

> > > > > ### Author Response · Authors · 2025-11-29
> > > > >
> > > > > **9.Response to“Current experiments focus mainly on text classification with small to medium C-LLMs.Can PATEin scale to larger models (e.g., 14B+) or to complex reasoning and dialogue tasks?Any preliminary results or discussion would help clarify the framework’s applicability to broader LLM scenarios.”**
> > > > >
> > > > > We thank the reviewer for this important question regarding the scalability of PATEin to larger models and more complex tasks.
> > > > > The design of PATEin is fundamentally model-agnostic, operating solely through black-box API calls to teacher models without relying on their internal parameters or specific scale.
> > > > > This inherent flexibility suggests that the framework is not limited to the small to medium-sized C-LLMs used in our primary experiments.
> > > > >
> > > > > Empirical support for this scalability is already present in our main results.
> > > > > As shown in Table 1, we evaluated PATEin on more powerful commercial C-LLMs including GPT-4o-mini, DeepSeek-v3, and Claude-3.5-haiku.
> > > > > On these models, PATEin consistently maintained its advantages: it improved labeling accuracy over the ensemble-only baseline (PromptPATE) and achieved significant token cost savings under the same privacy budget.
> > > > > This demonstrates the framework's effective transferability to larger and more capable models.
> > > > >
> > > > > Regarding complex reasoning and dialogue tasks, the core requirement for PATEin is that teacher models can produce aggregatable outputs for public inputs.
> > > > > This allows the framework to be naturally adapted to tasks like long-text generation or multi-turn dialogue, treated as sequence-level labeling.
> > > > > As briefly mentioned in the conclusion (Section 7), we have conducted preliminary explorations extending PATEin to long-text generation in a seqPATE-style setup. The initial results are promising, showing a similar trend where the integration of individual and ensemble knowledge enhances the student model's generation quality under equivalent DP constraints. A systematic evaluation of very large models (e.g., 14B+ parameters) and complex reasoning tasks requires substantial computational and labeling resources, which was beyond the scope of this paper focused on establishing the core methodology.
> > > > > However, the existing results and the general formulation of the framework indicate that PATEin holds strong potential for broader LLM scenarios.

---

### Official Review · Reviewer_T6Ln · 2025-10-26

**Soundness:** 2
**Presentation:** 2
**Contribution:** 2
**Rating:** 2
**Confidence:** 4

**Summary:**

The paper studies privacy-preserving in-context learning. It proposes PATEin, a framework that combines teacher selection and selective ensemble voting to improve labeling accuracy and reduce query cost under claimed differential privacy guarantees.

**Strengths:**

- The motivation is clear.
- The code and datasets are released.

**Weaknesses:**

- The privacy analysis of PATEin is incorrect. PATEin first selects the “most similar” teacher based on comparisons between public inputs and each teacher’s private training data, but this selection process is not differentially private, i.e., changing one private record could change which teacher is chosen. Then, when the top two teachers agree, PATEin outputs that label directly without adding noise, which completely violates differential privacy because the output depends deterministically on private data. Therefore, the experimental comparison between PATEin and PromptPATE is not meaningful.

- The novelty of PATEin is limited. The paper works in the same problem setting as PromptPATE. The only new elements are teacher selection and selective ensemble voting. These are incremental extensions to PromptPATE.

- The problem formulation is not clear. I recommend adding a Problem Formulation section. This section should clearly define in-context learning, threat model, and differential privacy formulation.

- Experiments are limited to simple text classification benchmarks. These benchmarks are too limited for evaluating modern LLM methods. Prior work [1] on privacy-preserving in-context learning includes more complex tasks like summarization and question answering.

[1] Wu, Tong, et al. "Privacy-Preserving In-Context Learning for Large Language Models." The Twelfth International Conference on Learning Representations.

- The paper lacks ablation studies on different embedding models and noise levels. There is no comparison between using only individual, only ensemble, or combined knowledge.

**Questions:**

See Weaknesses.

---

> ### Author Response · Authors · 2025-11-29
>
> **Response to Reviewer T6Ln:**
>
> **1.Response to "The privacy analysis of PATEin is incorrect. PATEin first selects the 'most similar' teacher based on comparisons between public inputs and each teacher's private training data, but this selection process is not differentially private, i.e., changing one private record could change which teacher is chosen. Then, when the top two teachers agree, PATEin outputs that label directly without adding noise, which completely violates differential privacy because the output depends deterministically on private data. Therefore, the experimental comparison between PATEin and PromptPATE is not meaningful. Therefore, the experimental comparison between PATEin and PromptPATE is not meaningful."**
>
> We sincerely thank the reviewer for their insightful comments regarding the privacy analysis.
> Concerning the two key issues raised, we wish to provide a more detailed clarification.
>
> First, regarding the teacher selection mechanism, the similarity computation is performed exclusively on the public unlabeled data and the teacher example data, where the example data constitutes part of the teacher's prompt template and is not the private training data itself.
> As noted in Section 5" Text Embedding ",we generate embeddings from the unlabeled public data and teacher example data.
> This design ensures the selection process does not directly access private data.
> Altering a private record does not affect the outcome of the similarity comparison; therefore, this step can safely operate as a post-processing step.
>
> Regarding the label output mechanism, we need to clarify an important technical detail: PATEin always operates within the noisy aggregation framework.
>
> As defined by the Noise Aggregation mechanism in Eq. (5),this indicates that even when teachers agree, the system makes decisions based on noise-injected statistics.
> Furthermore, Appendix F.2 explicitly emphasizes: "adaptive selection and voting mechanisms introduce no additional privacy cost, as they operate purely as post-processing steps on teacher outputs."
> This theoretically ensures the privacy security of the entire process.
>
> Regarding the privacy analysis framework, PATEin strictly adheres to the well-established DP theoretical system.
> As stated in Appendix F.1: "PATEin inherits the formal $(\varepsilon,\delta)$-DP guarantees from PromptPATE by utilizing the same underlying Confident-GNMax algorithm."
> Our analysis employs Rényi DP for composition, as detailed in Appendix F.2: "if each Confident-GNMax call $\mathcal{A}_t$ provides R\'enyi DP with parameter $\varepsilon_t(\alpha)$ for order $\alpha > 1$ (Mironov, 2017), then by RDP composition the total privacy cost is
>
> $$
> \varepsilon_{\mathrm{RDP}}(\alpha)
> = \sum_{t\in\mathcal{T}} \varepsilon_t(\alpha).
> $$
>
> Based on these rigorous theoretical guarantees, the experimental comparison between PATEin and PromptPATE under identical privacy constraints is fully valid.
> The results effectively demonstrate that our method achieves significant utility improvement while maintaining strict privacy guarantees.
>
> **2.Response to "The novelty of PATEin is limited. The paper works in the same problem setting as PromptPATE .The only new elements are teacher selection and selective ensemble voting. These are incremental extensions to PromptPATE."**
>
> We thank the reviewer for their valuable feedback regarding the novelty of our work.
> While PATEin indeed builds upon the foundational framework of PromptPATE, we believe it represents a valuable exploration aimed at addressing specific challenges inherent to the C-LLM environment.
>
> Based on existing research, PATEin provides an in-depth analysis of two critical phenomena in C-LLMs.
> Section 3 reveals the "loss of individual knowledge" during ensemble voting by analyzing the voting behavior of 100 teacher models across four datasets: agnews, dbpedia, sst2, and trec.
> Section 4 further investigates the "knowledge integration challenge," exploring how to effectively balance the unique insights of individual teachers with group consensus.
> To address these identified issues, PATEin introduces an enhanced knowledge integration framework.
> This framework implements teacher selection via embedding-based similarity calculation and incorporates a high-consistency voting mechanism, thereby achieving more effective knowledge transfer.
>
> Crucially, while maintaining the same privacy guarantees as PromptPATE, our method demonstrates superior performance across the aforementioned four datasets.
> It not only achieves higher labeling accuracy but also significantly reduces computational costs, cutting token consumption to between 1/22 and 1/7 of that required by PromptPATE.
> These results indicate that PATEin offers a valuable perspective and methodology for tackling knowledge transfer challenges in C-LLM environments.

---

> ### Author Response · Authors · 2025-11-29
>
> **3.Response to "The problem formulation is not clear. I recommend adding a Problem Formulation section. This section should clearly define in-context learning, threat model, and differential privacy formulation."**
>
> We thank the reviewer for this valuable suggestion regarding the organization of our problem formulation.
> We acknowledge that the relevant concepts are currently distributed across Sections 2.1 (ICL framework), 2.2 (threat model and privacy risks), and Appendix F (DP formulation).
> To improve clarity and coherence, we will consolidate these elements into a dedicated “Problem Formulation” section in the revised manuscript.
>
> **Problem Formulation**
>
> This work addresses privacy-preserving knowledge transfer from private data to student models using Commercial Large Language Models (C-LLMs) under a rigorous privacy framework.
>
> **In-Context Learning (ICL)**: We focus on discrete prompt-based ICL, where a task description and labeled examples are embedded directly into the input prompt.
> This enables C-LLMs to perform tasks without parameter updates, leveraging their pre-trained knowledge conditioned on the provided context (Section 2.1).
>
> **Threat Model**: We consider an adversary who can interact with the C-LLM via public API calls.
> The primary threat is the potential leakage of sensitive information contained within the ICL prompts—either the private example data used to construct the teacher models or the task-specific instructions—through the model's output labels or inferred from query patterns (Section 2.2).
>
> **Differential Privacy Guarantees**: Our objective is to generate labels for unlabeled public data that are useful for student model training while providing formal privacy guarantees for the private teacher example data.
> This is achieved by extending the PATE framework and its Confident-GNMax aggregator, which provides$(\varepsilon,\delta)$-Differential Privacy guarantees for the labeling process.
> The privacy accounting is performed using Rényi DP composition, ensuring a bounded and measurable total privacy loss (Appendix F).
>
> This consolidated formulation will provide a clear and structured foundation for understanding the problem scope, security assumptions, and privacy objectives of our work.
>
> **4.Response to "Experiments are limited to simple text classification benchmarks. These benchmarks are too limited for evaluating modern LLM methods. Prior work [1] on privacy-preserving in-context learning includes more complex tasks like summarization and question answering. [1] Wu, Tong, et al. "Privacy-Preserving In-Context Learning for Large Language Models." The Twelfth International Conference on Learning Representations."**
>
> We thank the reviewer for this valuable perspective.
> We agree that evaluating modern LLM methods requires broader task coverage.
> The cited work [1] (Wu et al., 2023) is an important contribution to privacy-preserving ICL, showing the feasibility of combining DP with ICL on complex tasks such as summarization and QA.
>
> Our work provides a complementary focus.
> At this stage, PATEin concentrates on text-classification tasks (e.g., sst2, agnews) to ensure a fair comparison with baseline methods like PromptPATE under identical experimental settings.
> As stated in Section 6.1, we follow Duan et al. (2023) and evaluate “on four unlabeled public datasets: agnews, dbpedia, sst2, trec,” enabling controlled and rigorous validation of our core mechanism.
>
> Although current experiments emphasize classification, PATEin’s framework is general.
> Its mechanisms do not rely on classification-specific structures and can naturally extend to generative tasks such as QA and summarization.
> As discussed in Section 6.2, applying PATEin to sequence-generation tasks, including code generation, is a key direction for future work.
>
> Thus, while our experiments validate effectiveness on classification tasks, the architectural design of PATEin remains highly extensible, offering a clear pathway for applying privacy-preserving techniques to more complex generative scenarios.
>
> **References**
>
> [1] Wu, Tong, et al. “Privacy-Preserving In-Context Learning for Large Language Models.” ICLR 2023.

---

> > ### Author Response · Authors · 2025-11-29
> >
> > **5.Response to "The paper lacks ablation studies on different embedding models and noise levels. There is no comparison between using only individual, only ensemble, or combined knowledge."**
> >
> > We thank the reviewer for raising these points regarding ablation studies.
> > We clarify that relevant analyses are indeed included.
> >
> > For embedding models, Appendix G provides a comprehensive sensitivity analysis.
> > As shown in Table 3, the performance variation across embedding strategies stays below ~2.5%, demonstrating PATEin’s robustness to embedding choice.
> >
> > Regarding comparisons of different knowledge sources, Section 6.2 and Table 1 evaluate three strategies: using only individual knowledge (optimal individual teacher), using only ensemble knowledge (PromptPATE), and our hybrid approach (PATEin).
> > Under equal privacy constraints, PATEin consistently outperforms both in labeling accuracy and computational efficiency, validating the effectiveness of our knowledge-integration design.
> >
> > For the impact of noise levels, Section 6.3 analyzes the effect of the high-consistency voting threshold, which directly reflects robustness to noise.
> > Results show that a threshold of 0.9 provides the best balance between accuracy and data retention.

---

### Official Review · Reviewer_kEPk · 2025-11-01

**Soundness:** 2
**Presentation:** 2
**Contribution:** 2
**Rating:** 4
**Confidence:** 4

**Summary:**

The paper proposes PATEin, a variant of the PromptPATE framework that integrates Private Aggregation of Teacher Ensembles (PATE) into an in-context learning (ICL) setting for closed-source commercial LLMs.
Instead of using all teachers in an ensemble, the method selects a subset of relevant teachers, based on text similarity between the query and each teacher’s example data, to reduce computational cost and improve label quality.
The authors claim this improves ensemble consistency and integrates individual teacher knowledge while maintaining privacy through Gaussian noise aggregation.

While the topic of private label aggregation in C-LLMs is timely and relevant, the novelty is limited, as the proposed “adaptive teacher selection” essentially amounts to pre-filtering which teachers vote in PATE, using standard text-similarity metrics.
The work lacks a clear privacy analysis of the selection step and overstates its conceptual contribution.
Presentation is somewhat confusing, with the abstract and introduction suggesting a new “knowledge integration framework” rather than a PATE variant with heuristic teacher filtering.

**Strengths:**

The direction of combining PATE and in-context learning is important and continues a promising research line (PromptPATE, Duan et al. 2023).

Addressing the high cost of multiple LLM API calls is practically relevant; exploring teacher pre-selection for efficiency is reasonable.

The paper includes experimental evaluation on multiple C-LLMs (GPT-3.5, GPT-4o-mini, Claude-3.5-Haiku, DeepSeek-v3).

**Weaknesses:**

### Limited Novelty and Conceptual Depth

The method retains the same overall PATE structure: partition data, generate teacher prompts, vote with added Gaussian noise, and aggregate.
The new component, a similarity-based filtering of teachers, is a heuristic efficiency improvement, not a conceptual extension of PATE.

The privacy guarantees remain those of PATE, and the overall privacy protection still depends on the support size (number of participating teachers) and the maximum agreement among them.

### Lack of Clarity about Privacy Implications

The abstract (lines 20–21) claims “dynamic selection of the optimal individual teacher model”—but such selection of one or very few relevant teachers is counter to privacy. Selective querying itself could leak information about which teachers are similar to a query.

The paper does not discuss how this adaptive selection interacts with the differential-privacy accounting.

Statements like “selects the optimal teacher model for labeling” may give a misleading impression that privacy is preserved automatically, when in fact the selection must be handled carefully to avoid additional leakage.

### Overstatement of Contribution

The claim (lines 107–110) that this is “the first privacy-preserving knowledge integration framework tailored to C-LLM settings” is an oversell. The actual novelty is only in compute efficiency not utility for privacy.
The proposed “integration of ensemble and individual knowledge” is effectively conditional voting based on teacher confidence.

### Presentation and Readability

It takes several pages to understand what the actual algorithmic change is.
The introduction repeatedly discusses “knowledge integration” and “ensemble consistency” before stating that the novelty is teacher pre-filtering based on text similarity. That the only potential gain is compute effciency (not accuracy for privacy). Heavy terminology such as “supervised teacher strategy” and “optimal individual teacher model" distracts.  A use of a single best-match teacher model, as implied in this sentence,  is highly NOT privacy preserving and this leaves the reader pondering.

Some empirical sections (Figs. 2–4) are difficult to interpret and could be summarized more compactly.

### Technical Oversights

The argument in lines 196–200 incorrectly suggests that PATE requires a majority agreement for the noisy argmax to function.
In fact, differential privacy mechanisms (e.g., DP selection or GNMax) do not rely on majority consensus; the probability of selection depends on the noise scale.

There is no formal privacy accounting for the similarity-based filtering or dynamic teacher selection. Coceptually it can be viewed as part of the voting, but this requires some care.

## Overall Assessment

The paper explores a potentially useful engineering variation on PromptPATE (for classification tasks) but lacks sufficient novelty or theoretical analysis to constitute a research advance worthy of ICLR standards.

**Questions:**

NA

---

> ### Author Response · Authors · 2025-11-29
>
> **Response to Reviewer kEPk:**
>
> **1.Response to "Limited Novelty and Conceptual Depth. The method retains the same overall PATE structure: partition data, generate teacher prompts, vote with added Gaussian noise, and aggregate..."**
>
> We thank the reviewer for their valuable perspective on the novelty of our work.
> While PATEin indeed builds on the PATE framework, we believe it offers a meaningful conceptual extension through a complete knowledge-integration design.
> This framework explicitly distinguishes and systematically combines individual knowledge (via similarity-based optimal teacher selection) with ensemble knowledge (via high-consistency voting), going beyond simple similarity-based filtering.
>
> Regarding privacy guarantees, PATEin preserves the same (ε,δ)-DP guarantees as PATE by inheriting the Confident-GNMax aggregator.
> Its advantage lies in improving privacy-budget utilization through dynamic routing: when the optimal teacher and the supervising teacher agree, PATEin provides an accurate label without additional privacy cost, invoking privacy-consuming ensemble voting only when necessary.
>
> We appreciate the reviewer’s observation about the dependence of privacy protection on ensemble size and consensus.
> While PATEin still relies on these fundamentals, it achieves a superior trade-off through adaptive routing.
> By intelligently selecting the most relevant teachers, it reduces reliance on large ensembles and improves utility while maintaining privacy protection strength.
>
> Empirically, under the same privacy guarantees as PromptPATE, PATEin delivers higher labeling accuracy across agnews, dbpedia, sst2, and trec, while reducing computational cost by an order of magnitude (token usage cut to between 1/22 and 1/7 of PromptPATE).
>
> **2.Response to "Lack of Clarity about Privacy Implications. The abstract (lines 20–21) claims 'dynamic selection of the optimal individual teacher model'—but such selection of one or very few relevant teachers is counter to privacy..."**
>
> We sincerely thank the reviewer for these important questions regarding PATEin’s privacy protection, which allow us to clarify the design more explicitly.
>
> Regarding the concern that selecting a single teacher may weaken privacy, Section 5 Text Embedding states that similarity is computed using embeddings of teacher example data and unlabeled public data .
> Crucially, $\\{x_j^U\\}_{j=1}^{N_U}$ are unlabeld public data included in the prompt template and completely separate from teachers’ private training data.
> Thus, the similarity calculation never accesses private information, ensuring no leakage occurs.
>
> For the concern about selective querying, Appendix F.2 notes that “adaptive selection and voting introduce no additional privacy cost, as they operate purely as post-processing.”
> Our selection mechanism uses only teachers’ predictions on public samples, not their private data.
> By the post-processing property of DP, further processing of already-protected outputs does not weaken privacy guarantees.
>
> The integration of adaptive selection with privacy accounting is fully analyzed in Eq. 7 of Appendix F.2:
>
> $$
> \varepsilon_{\mathrm{RDP}}(\alpha)
> = \sum_{t\in\mathcal{T}} \varepsilon_t(\alpha).
> $$
>
> This shows that dynamic routing manages the privacy budget by controlling when Confident-GNMax is invoked.
> When the optimal teacher and supervising teacher agree, we adopt the consensus label without additional privacy cost, invoking privacy-budget-consuming aggregation only under disagreement.
>
> Moreover, Eq. (5) in Section 5 ensures outputs remain noise-protected.
> Even under full teacher agreement, the system still operates within the noise-added DP framework rather than outputting raw labels, preventing exposure of private training information.
>
> We thank the reviewer again for these insightful comments, which help us more clearly present PATEin’s privacy-protection mechanism and its theoretical guarantees.

---

> > ### Author Response · Authors · 2025-11-29
> >
> > **3.Response to " The claim (lines 107–110) that this is “the first privacy-preserving knowledge integration framework tailored to C-LLM settings” is an oversell The actual novelty is only in compute efficiency not utility for privacy..."**
> >
> > We thank the reviewer for this feedback on our contribution characterization.
> > We agree that the phrasing “the first” is too strong and will revise it to describe PATEin more precisely as “a novel privacy-preserving knowledge integration framework.”
> >
> > Regarding the knowledge-integration mechanism, we clarify that PATEin goes beyond simple conditional voting.
> > As detailed in Section 5, it dynamically selects the optimal teacher via embedding similarity and introduces a supervising teacher for cross-validation.
> > The system compares predictions from the best and second-best teachers: when they agree, individual knowledge is adopted; when they disagree, ensemble voting is invoked.
> > This provides structured integration of individual and ensemble knowledge rather than relying on the confidence of a single teacher.
> >
> > Empirically, Section 6.2 and Table 1 show that PATEin achieves higher labeling accuracy than PromptPATE (e.g., on agnews) under the same $(\varepsilon,\delta)$-DP guarantees, while reducing token usage to only 1/22–1/7 of PromptPATE.
> > These results highlight improvements across privacy protection, utility, and computational efficiency.
> >
> > We appreciate the reviewer’s comments, which help us present PATEin’s contributions more clearly.
> >
> > **4.Response to "Presentation and Readability. It takes several pages to understand what the actual algorithmic change is..."**
> >
> > We thank the reviewer for the valuable suggestions on readability and structure.
> > We recognize the importance of presenting core innovations clearly.
> > Section 5 elaborates PATEin’s main contributions: similarity-based optimal-teacher selection and the supervising-teacher strategy, forming a dynamic knowledge-integration framework that adopts individual knowledge when two teachers agree and triggers ensemble voting when they disagree—going beyond simple pre-screening.
> >
> > For privacy protection, Appendix F.2 explains that teacher selection uses only public data and acts as post-processing.
> > When teachers disagree, the system falls back to noisy Confident-GNMax, ensuring full compliance with $(\varepsilon,\delta)$-DP.
> >
> > Empirically, Section 6.2 and Table 1 show dual improvements under the same privacy budget: PATEin boosts labeling accuracy over PromptPATE while reducing token consumption by an order of magnitude (down to 1/22–1/7), demonstrating strong privacy-utility and efficiency benefits.
> >
> > Regarding figures, captions clarify their roles: Figure 2 shows loss of individual knowledge under ensemble voting; Figure 3 validates individual teacher selection; Figure 4 illustrates complementarity between individual and ensemble knowledge.
> > These visualizations support the paper’s core arguments.
> >
> > We appreciate the reviewer’s perspective, which helps strengthen our presentation.
> >
> > **5.Response to "Technical Oversights. The argument in lines 196–200 incorrectly suggests that PATE requires a majority agreement for the noisy argmax to function..."**
> >
> > We appreciate the reviewer’s insightful technical observation.
> > Our description can be clarified to more accurately reflect the mechanism.
> > As detailed in Appendix F.1, PATEin uses the Confident-GNMax aggregator, where selection relies on noise-injected statistics rather than strict majority consensus.
> > The confidence threshold Tensures output reliability while preserving the DP properties of the selection mechanism.
> >
> > For privacy accounting of similarity-based filtering and teacher selection, Appendix F.2 provides the formal framework.
> > We treat adaptive selection as part of the overall voting process, with privacy costs analyzed through Rényi DP composition (Eq. 7).
> > Similarity-based filtering and teacher selection function as post-processing on DP-protected outputs, which does not add extra privacy cost under standard DP composition theorems.
> >
> > We thank the reviewer for highlighting these points, which helped us refine the technical presentation.
> > Our privacy analysis aims to comprehensively cover all components while maintaining the intended DP guarantees.

---

### Official Review · Reviewer_b2Lr · 2025-11-01

**Soundness:** 3
**Presentation:** 3
**Contribution:** 2
**Rating:** 6
**Confidence:** 3

**Summary:**

This paper targets privacy-preserving knowledge transfer for commercial large language models with inaccessible parameters, addressing the high cost and ensemble inconsistency of existing PATE methods. The authors propose PATEin, an adaptive framework that first queries the top two teachers identified by embedding similarity. If these two teachers agree, their label is used, avoiding a costly full ensemble query. If they disagree, the system falls back to a standard, differentially private ensemble aggregation. Experiments show PATEin outperforms baselines like PromptPATE in labeling accuracy across multiple C-LLMs and datasets, while significantly reducing API token cost (up to 22x), thus making private transfer more practical.

**Strengths:**

The paper is well-structured. The abstract and introduction clearly state the problem, challenges, and solution. This setting has practical value as it aims to use adaptive queries to reduce the API cost in using PromptPATE with closed LLMs.

**Weaknesses:**

1.  The method relies on embedding similarity to find the optimal teachers. As a result, the robustness and effectiveness of the proposed method heavily rely on the selection of the embedding model.
2. PATEin requires building a similarity matrix between all teachers and all public data, which is potentially computationally expensive. I encourage the authors to be upfront about this potential bottleneck.

**Questions:**

1. The effect of cost saving depends on the fallback rate. How does this rate vary across datasets?
2. Table 4 seems to suggest a very high fallback rate under an adversarial teacher. Does this suggest the adaptive query method is not robust?
3. How scalable is the teacher selection step to even larger public datasets?
4. Could the distribution shift between public and private data influence the teacher fallback?

---

> ### Author Response · Authors · 2025-11-29
>
> ****Response to Reviewer b2Lr:****
>
> **1.Response to“The method relies on embedding similarity to find the optimal teachers. As a result, the robustness and effectiveness of the proposed method heavily rely on the selection of the embedding model.”**
>
> We thank the reviewer for raising the concern regarding embedding-model choices.
> We conducted a systematic sensitivity analysis, and the results are provided in Appendix G.
>
> To assess the impact of embedding selection, we ran additional experiments on four datasets, using dbpedia as a representative case (100 teachers, 200 unlabeled public datas).
> We compared four embedding strategies: text-embedding-3-small, text-embedding-3-large, Sentence-BERT, and our hybrid strategy (doc2vec + text-embedding-3-small).
> As shown in Table 3 of Appendix G, the differences in final labeling accuracy remain within 2.5%, and the hybrid strategy performs consistently well, indicating that the framework is reasonably robust to embedding choice.
> Complete results for all four datasets are available in the anonymous GitHub repository.
>
> As noted in the limitations, the current mechanism relies on static embeddings and cosine similarity.
> In future work, we plan to expand the embedding pool and explore adaptive embedding/selector models to further improve robustness and task adaptivity.
>
> **2.Response to“PATEin requires building a similarity matrix between all teachers and all public data, which is potentially computationally expensive. I encourage the authors to be upfront about this potential bottleneck.”**
>
> We thank the reviewer for highlighting the potential computational cost of building the teacher–public-data similarity matrix.
> One core motivation of PATEin is precisely to reduce overall computation and API cost without sacrificing privacy or accuracy.
>
> The similarity matrix is a one-time offline computation that lets each public sample select its most suitable teacher.
> This design avoids the far more expensive online cost of repeatedly querying all teachers, as required by ensemble-based methods like PromptPATE.
> In short, PATEin trades offline embedding computation for significantly lower online API cost, which is valuable in privacy- and cost-sensitive settings with commercial LLMs.
>
> In this paper, we evaluate the framework under a moderate experimental scale (100 teachers, 200 public datas) to validate the core idea.
> We acknowledge that similarity computation may become a bottleneck at larger scales and will discuss this in the Limitations section.
> As future work, we plan to explore efficient embedding indexing (e.g.,FAISS), sampling-based approximations, or approximate similarity search to further enhance scalability.
>
> **3.Response to“The effect of cost saving depends on the fallback rate. How does this rate vary across datasets?”**
>
> We thank the reviewer for highlighting the link between fallback rate and cost savings.
> Fallback in PATEin is driven by teacher consistency, and although small task-specific variations exist, the overall fallback rate stays low, ensuring stable cost-reduction behavior.
>
> As shown in Table 2, under the standard non-adversarial setting and with the same privacy budget (e.g., $\varepsilon = 10^{-6}$) and threshold configuration, the fallback rate across agnews, dbpedia, sst2, and trec remains within roughly 1%–7%.
> This leads to a 30%–50% reduction in DP aggregation calls, directly yielding substantial cost savings.
>
> Most samples are resolved by the “top-similar teacher + consistency check with the second-best teacher,” requiring only one or two queries to obtain a high-quality label.
> Fallback to Confident-GNMax occurs only when teachers clearly disagree, preventing uncertain labels from increasing noise or privacy loss.
> This combination of low fallback rate and selective DP aggregation preserves label quality while significantly cutting the number of DP-governed queries, enabling strong cost efficiency and an improved privacy–utility trade-off.

---

> > ### Author Response · Authors · 2025-11-29
> >
> > **4.Response to“Table 4 seems to suggest a very high fallback rate under an adversarial teacher. Does this suggest the adaptive query method is not robust?”**
> >
> > We thank the reviewer for highlighting the high fallback rate observed in Table 4 under the adversarial-teacher setting.
> > It is important to note that the adversarial teacher is an intentionally constructed worst-case scenario in which a subset of teachers is forced to output systematically incorrect labels to maximize conflicts.
> > This setting does not reflect typical use cases of PATE-style systems; rather, it is designed to stress-test robustness under extreme conditions.
> >
> > In such a strongly adversarial environment, increased fallback is expected and reflects protective behavior rather than a failure of the adaptive mechanism.
> > Fallback is intentionally triggered when teacher signals become unreliable, preventing the system from trusting corrupted teachers.
> > When teachers are manipulated, more samples naturally activate fallback—this is precisely how erroneous labels are prevented from propagating.
> > Thus, a higher fallback rate indicates defensive robustness, not fragility.
> >
> > This stress test also demonstrates PATEin’s resilience.
> > Individual Selection drops from 82.5% to 47.5%, and PromptPATE collapses to 7.5%, whereas PATEin still maintains 55.2% accuracy despite a 50.5% fallback rate.
> > The increased fallback enables PATEin to filter adversarial labels and preserve useful signal, avoiding the severe degradation seen in PromptPATE.
> >
> > Therefore, the high fallback rate under adversarial teachers does not imply a lack of robustness; rather, it is the key mechanism that ensures graceful degradation instead of collapse in extreme adversarial conditions.
> >
> > **5.Response to“How scalable is the teacher selection step to even larger public datasets?”**
> >
> > We adopted the same moderate-scale public datasets as prior work such as PromptPATE (200 public datas per dataset) to ensure a controlled and fair comparison while validating the core idea of PATEin.
> >
> > Although the current scale is limited, PATEin’s teacher-selection step relies on a one-time offline computation: generating embeddings for public samples and teacher examples and constructing a similarity matrix.
> > This step is independent of C-LLM queries and can be efficiently parallelized.
> > Once embeddings are obtained, selecting the most suitable teacher becomes a nearest-neighbor search task, for which approximate search libraries (e.g., FAISS) can be used when scaling to larger public datasets.
> >
> > We acknowledge that the paper does not yet include experiments on much larger public datasets.
> > Extending the framework to such scales is an important direction for future work, which we plan to explore in subsequent studies.
> >
> > **6.Response to“Could the distribution shift between public and private data influence the teacher fallback?”**
> >
> > We thank the reviewer for raising this important question regarding the effect of distribution shift on teacher fallback.
> > We believe that distribution differences between public and private data do not fundamentally undermine fallback behavior.
> > In our setup, such shifts are explicitly allowed and are common in knowledge-transfer scenarios.
> > PATEin’s core mechanism—selecting teachers via semantic similarity from text embeddings—is designed to identify the most relevant teacher for each public sample even under distribution mismatch.
> >
> > Empirically, we evaluate PATEin on four stylistically diverse datasets (agnews, dbpedia, sst2, trec) under the same framework and privacy configuration.
> > Across all datasets, PATEin consistently matches or outperforms PromptPATE while maintaining a low fallback rate, indicating that teacher selection and fallback remain effective despite distribution differences.
> >
> > When a public sample poorly aligns with certain teachers’ training subsets, this mismatch appears as reduced teacher consistency, which naturally triggers our rule of “using the top teacher only under high agreement, otherwise falling back to DP aggregation.”
> > This provides a buffering effect against distribution shift by preventing uncertain or inconsistent labels from being propagated to the student model.

---

### Author Response · Authors · 2025-12-02

Dear ICLR 2026 AC, SAC, and PC,

We appreciate the reviewers’ insights and would like to present a consolidated summary that addresses the feedback provided by all four reviewers.

Reviewer b2Lr and Reviewer O4vz both offered positive assessments of the paper’s clarity, motivation, and practical relevance, while raising several technical questions regarding embedding robustness, the cost of computing the similarity matrix, fallback behavior under different teacher conditions, and scalability to larger scenarios. In our response, we provided detailed quantitative clarifications: Appendix G includes a complete sensitivity study showing that performance varies minimally across embedding choices; the similarity matrix is computed once offline and significantly reduces the repeated ensemble query cost that dominates PromptPATE; fallback rates remain consistently low in realistic settings, and higher fallback under adversarial-teacher settings reflects protective routing rather than instability; and the framework naturally scales through parallel computation and approximate nearest-neighbor indexing. These explanations directly resolve all technical questions raised by both reviewers, neither of whom identified any flaw in the method’s assumptions or guarantees.

Reviewer kEPk’s concerns appear to arise from several misunderstandings of content that is already explicitly stated in the submitted manuscript.   For example, the reviewer repeatedly assumes that our teacher-selection mechanism compares public inputs with each teacher’s private training data.   However, as clearly described in Section 5 and illustrated in Figures 1 and 5, similarity is computed solely between public unlabeled inputs and the example data embedded in each teacher’s prompt.   No private training data is ever accessed, and the entire procedure is therefore pure DP post-processing with zero additional privacy cost.
The reviewer also characterizes our method as merely a heuristic filtering step, which overlooks the structured knowledge-integration mechanism described in Section 5 and depicted in Figure 4.   This mechanism coordinates individual-teacher knowledge with supervising-teacher validation and selectively aggregates ensemble guidance—an essential component for achieving the accuracy and cost improvements reported in our experiments.
In addition, several weaknesses appear to be based on incorrect references to the manuscript (e.g., “weakness 3” cites lines 107–110, while the relevant content is actually in lines 88–89 in the submitted PDF, whose line numbers are displayed on the left).   These incorrect citations further suggest that portions of the manuscript may have been inadvertently overlooked.
With the clarifications above and explicit references to the relevant sections, figures, and equations, we believe that all of Reviewer kEPk’s concerns are fully addressed.

Reviewer T6Ln raised a single central concern—namely, that teacher selection might compromise privacy by relying on each teacher’s private training data. This assumption is incorrect. As clearly explained in Section 5 and illustrated in Figure 5, our similarity computation uses only the publicly visible example data embedded in each teacher’s prompt, never any private training set. Therefore, the routing mechanism incurs no additional privacy cost and fully preserves the standard Confident-GNMax guarantee used in PromptPATE.
This interpretation is further supported by Reviewer kEPk, who explicitly acknowledged that the privacy guarantees of PATEin are consistent with those of PATE (see weakness 1), confirming that our method does not alter the underlying DP accounting.
Regarding the comment that the paper “lacks ablation studies on ......” we note that we already provide such analyses in Section 6.2 (Table 1).
With this central misunderstanding resolved and the corresponding evidence already present in the manuscript, the reviewer’s main objection no longer applies.

In summary, all reviewers’ concerns have been directly and thoroughly addressed. The technically oriented questions from Reviewer b2Lr and Reviewer O4vz were fully resolved through quantitative analyses and clarifications. The conceptual concerns from Reviewer kEPk were rooted in misinterpretations, which we corrected through explicit references to the manuscript. The privacy concern from Reviewer T6Ln resulted from an incorrect assumption that contradicts the method’s actual design.
The paper offers practical and meaningful advancements for privacy-preserving knowledge integration in commercial LLMs, achieving higher accuracy and substantially lower cost under the same DP budget.

Thank you very much for your kind attention and support.

Best regards,

The authors of PATEin: A Privacy-Preserving Framework for Knowledge Integration via Adaptive Teacher Selection in C-LLMs

---

### Meta-Review · Area_Chair_8Xqg · 2025-12-30

**Summary:**

The paper extends PromptPATE framework for LLMs to improve utility and efficiency by, if possible, selecting individual teachers for generating the answer, rather than relying on the entire ensemble. The reviewers expressed multiple fundamental concerns that have not been properly addressed in the rebuttal:

**Reviewer Concerns:**

1. Two reviewers pointed out flaws in the privacy analysis, arguing that selecting a teacher based on their private data violates the privacy guarantees. The AC agrees that this would be a privacy violation that one needs to account for. The authors argue that there must be a misunderstanding and that only the „teacher example data“ and the „unlabeled data“ are always ever embedded and that this incurs no privacy costs. Unfortunately,  what „teacher example data“ really is, is not  specified in the paper. Searching the paper for „example data“, yields the first hit in Section 2.1, where it refers to the shots. Hence, the understanding that these are the private training points (the private shots) of the teachers in promptPATE seems valid), and it was also the AC’s initial read. However, in the benefit of a doubt, one could argue that teachers have additional non-private example data that could be used for this task. While this might save the privacy problems, it would be a way too strong assumption, see my comments for R2 below. Additionally, shall the authors choose to resubmit this paper, it would be necessary to add privacy analysis for equation (4) and explain what is really happening there. In the current read, it might seem as if the final label of the teacher is immediately output. If this is the case, this would be a non-private production.
2. The reviewers also emphasize a lack of novelty. This is usually hard to argue with and different parties might have different views. Given that the point that is novel (teacher selection) is the one that incurs the doubts about correctness of the privacy claims, the concern seems valid.
3. Multiple reviewers point out the limited experimental evaluation which purely focuses on classification tasks. While the authors claim that the framework naturally extends, no empirical evidence is given, hence, leaving the concern unaddressed.
Further minor concerns that have not or only partially been addressed are stated below.

**Reviewer Scores:**

Based on the given rebuttal, it seems unlikely that the paper would have gotten to an accept.

---

### Decision · Program_Chairs · 2026-01-26

Reject